# Modularity of Zorya defense systems during phage inhibition

Giuseppina Mariano [1,2,6] ✉, Justin C. Deme [3,6] ✉, Jennifer J. Readshaw [4], Matthew J. Grobbelaar [4], Mackenzie Keenan[1], Yasmin El-Masri[1], Lindsay Bamford[1], Suraj Songra[1], Tim R. Blower [4], Tracy Palmer [5] & Susan M. Lea [3] ✉

Bacteria have evolved an extraordinary diversity of defense systems against bacteriophage (phage) predation. However, the molecular mechanisms underlying these anti-phage systems often remain elusive. Here, we provide mechanistic and structural insights into Zorya phage defense systems. Using cryo-EM structural analyses, we show that the Zorya type I and II core components, ZorA and ZorB, assemble in a 5:2 complex that is similar to inner-membrane ion-driven, rotary motors that power flagellar rotation, type 9 secretion, gliding and the Ton nutrient uptake systems. The ZorAB complex has an elongated cytoplasmic tail assembled by bundling the C-termini of the five ZorA subunits. Mutagenesis demonstrates that peptidoglycan binding by the periplasmic domains of ZorB, the structured cytoplasmic tail of ZorA, and ion flow through the motor is important for function in both type I and II systems. Furthermore, we identify ZorE as the effector module of the Zorya II system, possessing nickase activity. Our work reveals the molecular basis of the activity of Zorya systems and highlights the ZorE nickase as crucial for population-wide immunity in the type II system.

Bacteriophages (phages) co-exist with and predate bacteria in every environmental niche. Bacteria have evolved a plethora of diverse defense strategies to prevent viral infection[1–16], with phages co-evolving anti-defense counter-measures[3,17]. Historically, the characterization of restriction-modification systems (RM) and CRISPR-Cas fostered a revolution in gene editing and biotechnology[18,19]. More recently, it has emerged that bacteria and archaea harbor a much wider variety of defense systems[1–16], frequently encoded in chromosomal hotspots defined as 'defense islands'[2,10,11,20–22]. Since 2018, ~200 anti-phage systems have been reported, but a mechanistic understanding of their mode of action is available for only a few[1–16,23–26]. To date, mechanisms including depletion of NAD+ and of the cellular NTP pool, pore-formation, and bacterial DNA damage have been reported,

causing host cell death or stasis. This limits phage spread among the bacterial population, a phenomenon termed population-wide immunity or abortive infection[27,28]. In other instances, defense systems act by quickly sensing and degrading invading phage DNA, without affecting the survival of infected cells (also known as direct defense or first-line defense)[20,29,30].

The Zorya phage defense system was first discovered in 2018[2]. Initially described as two related systems, Zorya I and II, a third sub-type, Zorya III, was defined more recently[31]. Importantly, all Zorya systems share two components, ZorA and ZorB, containing domains distantly related, respectively, to the MotA and MotB subunits of the bacterial flagellar motor[2] (Fig. 1a, b). Zorya I additionally comprises ZorC and ZorD, the latter harboring a predicted DEAD-box helicase

[1]Department of Microbial Sciences, Faculty of Health and Medical Sciences, University of Glasgow, Guildford, UK. [2]School of Infection and Immunity, University of Glasgow, Glasgow, UK. [3]Center for Structural Biology, Center for Cancer Research, National Cancer Institute, NIH, Frederick, MD, USA. [4]Department of Biosciences, Durham University, Durham, UK. [5]Microbes in Health and Disease Theme, Newcastle University Biosciences Institute, Newcastle University, Newcastle upon Tyne, UK. [6]These authors contributed equally: Giuseppina Mariano, Justin C. Deme. ✉e-mail: giusy.mariano@glasgow.ac.uk; justin.deme@nih.gov; susan.lea@nih.gov

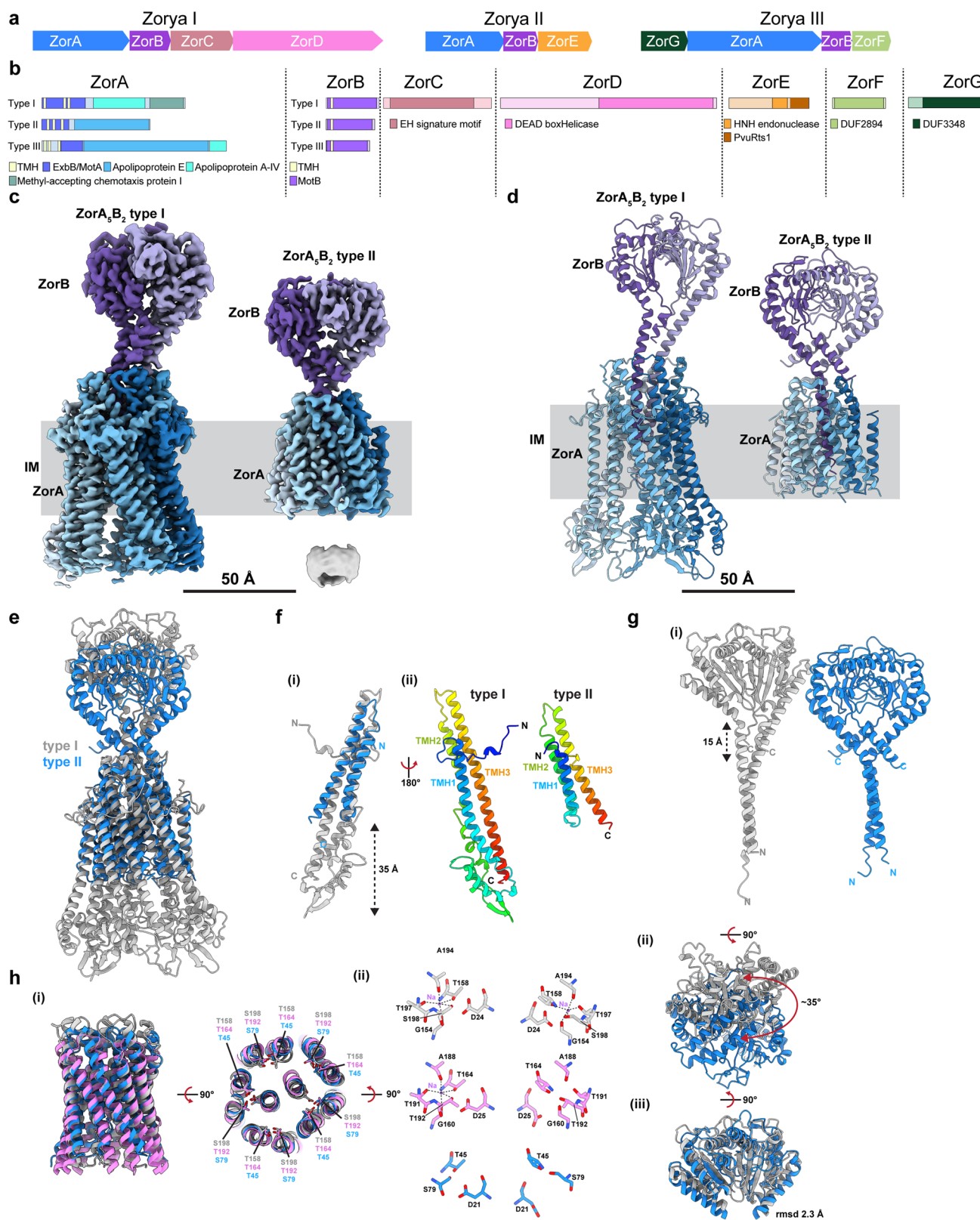

domain (Fig. 1a, b and Supplementary Fig. 1a, b). Zorya II harbors the ZorE component, a probable HNH endonuclease[2] (Fig. 1a, b and Supplementary Fig. 1c). Zorya III instead encodes a DUF3348 domain protein (ZorF) and a DUF2894 domain protein (ZorG)[31] (Fig. 1a, b).

Here we report the structures of ZorAB from the Zorya I and II systems, showing they assemble as 5:2 ion-driven motors as seen in flagellar, type 9, and Ton systems[32–34]. Unexpectedly, the ZorB

periplasmic domains are dimerized in these presumed inactive complexes and structurally resolved, unlike the equivalent region of the flagellar Mot complexes. This provides the first view of the arrangement of a peptidoglycan binding domain with respect to the inner membrane complex. The C-termini of ZorA bundle together to form a long, cytoplasmic extension. We demonstrate that both these elaborations of the core membrane complex are essential to elicit

**Fig. 1 | Structures of a ZorA₅B₂ type I complex from *Shewanella* sp. strain ANA-3 and a ZorA₅B₂ type II complex from *Sulfuricurvum kujiense*. a** Schematic representation of Zorya I, Zorya II, and Zorya III loci. **b** Schematic representation of the domain composition of each Zorya subtype component. Domains were predicted using HHPRED. Only predictions with an *e*-value < 0.01 were included. For sequences with multiple domains predicted, the highest scoring one (bit score) is reported. **c** Cryo-EM volumes of a type I ZorA₅B₂ complex from *Shewanella* sp. strain ANA-3 (left) and a type II ZorA₅B₂ complex from *Sulfuricurvum kujiense* (right). ZorA subunits are displayed in shades of blue and the centrally located ZorB subunits are in shades of purple. The inner membrane (gray) was assigned based on the micelle density surrounding the complex at lower contour levels. **d** Models of type I ZorA₅B₂ from *Shewanella* sp. strain ANA-3 (left) and type II ZorA₅B₂ from *Sulfuricurvum kujiense* (right) depicted as cartoon representations and colored as in (**a**). **e** Structural alignment of the type I (gray) and type II (blue) ZorA₅B₂ complexes shows a conserved core across both Zorya types. **f** (i) Structural alignment of a single ZorA subunit from type I (gray) and type II (blue) and (ii) side-by-side comparison of these subunits in the same orientation colored in the rainbow (N-terminus blue, C-terminus red). **g** Comparison of ZorB dimers from type I (gray) or type II (blue), from the same view as the overlay displayed in (**c**), demonstrates the peptidoglycan binding domain differs in height (i) and orientation (ii) but the overall fold is consistent across Zorya types (iii). **h** (i) Structural conservation of the ZorAB core (left) and the polar ring formed from conserved serines/threonines of the ZorA pentamer surrounding the critically conserved aspartates on ZorB is consistent across type I (gray), type II (blue) ZorAB and MotAB (PDB 8UCS; pink) complexes (right). (ii) The polar residues of *Shewanella* sp. strain ANA-3 ZorA coordinate sodium at both sites that recruit the conserved aspartate of ZorB (top, gray), whereas *C. sporogenes* MotAB recruits sodium at a single site (pink, middle) and no metal or ion coordination is observed in *Sulfuricurvum kujiense* ZorAB (blue, bottom).

protection against phages. For Zorya II, anti-phage activity requires the presence of the ZorE effector, which we show is recruited by ZorAB. Biochemical characterization indicates that ZorE is a nickase in vitro and prevalently mediates single-stranded breaks in the bacterial chromosome in vivo.

We conclude that Zorya systems, on their own, mediate protection by recruiting nuclease complexes that can damage host DNA in vivo, leading to population-wide immunity. In summary, our study uncovers the molecular mechanism underlying Zorya-mediated phage defense, revealing a highly sophisticated strategy to thwart phage infection.

## Results

### ZorAB is structurally related to MotAB but has Zorya-specific features

The three Zorya subtypes have a differential distribution across different bacterial strains (Supplementary Fig. 2a, b and Supplementary Data 1–3), typical of anti-phage systems. Zorya I is predominant in *Escherichia, Klebsiella, Vibrio, Pseudomonas,* and *Cronobacter* genera, whereas Zorya II is more abundant in *Escherichia* and *Campylobacter*. Zorya III is more frequently found in *Burkholderia, Xanthomonas, Stenotrophomonas, Ralstonia*, and *Paraburkholderia* (Supplementary Fig. 2a, b).

To investigate the mechanisms of Zorya systems we first targeted the structures of the Zorya core components ZorA and ZorB, focusing on Zorya I and Zorya II. ZorA I-ZorB I from *Shewanella sp. ANA-3* and ZorA II-ZorB II from *Sulfuricurvum kujiense* DSM 16994 yielded protein samples (Supplementary Fig. 2c, d) that allowed structure determination by cryoEM at global resolutions of 2.2 and 2.8 Å, respectively (Fig. 1 and Table 1). The structures reveal that both complexes consist of five copies of ZorA packed around two copies of the ZorB N-terminal helices, as seen in other 5:2 ion-driven motor complexes (Fig. 1c, d). The type I and II complexes largely differ in the degree of elaboration of ZorA and the length of the ZorB N-terminal helix, with the type I complex substantially larger than the type II (Fig. 1e, f, and g). This difference is clearly a characteristic of the different Zorya families as the size of the components is well-conserved within each family. Overlay of the intramembrane core of the complexes with the same region from the flagellar MotAB complex (Fig. 1h(i)) reveals that the core residues involved in ion-transduction are conserved, with the critical ZorB Aspartate residue (D24 (ZorB-type I); D21 (ZorB type II)) packing against a ring of serine/threonines presented by the surrounding ZorA subunits. Further analysis of the environment around the Asp suggests that Zorya systems, as for Mot systems, can be driven by the flow of different ions with our type I volume revealing Na+ ions bound to the conserved serine/threonine residues, as seen in the Na+ driven MotAB complex (Fig. 1h(ii)), whilst the environment in our type II model is more similar to that seen in proton-driven motors that lack an additional polar residue required for sodium coordination.

Unexpectedly, the C-terminal, presumed peptidoglycan-binding domains of the ZorB components, are ordered at the top of the N-terminal helices with the domains dimerized (Fig. 1). Similar domains are present at the C-terminus of MotB, but structures of the MotAB complex have not resolved these domains, and they are considered to be mobile with respect to the intramembrane complex in the inactive motor state. Although the arrangement of the ZorB peptidoglycan (PG) binding domain dimer with respect to the rest of the complex differs between the two Zorya systems (Fig. 1h(iii)) by a ~35° rotation, the dimer structure is well conserved (Fig. 1h(iii)). The C-terminal extension of the ZorA subunits (residues 236–696 of type I *Shewanella* ANA-3, residues 100–378 of type II *S. kujiense*) could not be fully resolved in these high-resolution volumes presumably due to its variable location with respect to the rest of the complex.

As for all other such complex structures solved to date, both complexes are presumed to be in an inactive state since they show no open path for ion flow from the periplasmic to the cytoplasmic face. In the MotAB complex, a region of MotB immediately above the membrane forms a 'plug' helix which folds back between the periplasmic MotA loops to lock the complex in a blocked state—activation is proposed to be driven by pulling this plug away from the membrane following engagement of the PG-binding domains. By comparison, the Zorya complexes are blocked via a collar formed from periplasmic extensions of ZorA TM helices 2 and 3 and elaborate ZorA periplasmic loops that pack against the ZorB subunit (as also seen in type 9 and Ton systems).

We next screened a panel of coliphages against *E. coli* MT56 expressing Zorya I or Zorya II from different bacteria (Supplementary Fig. 2e). We observed that homologs of Zorya I from *Serratia marcescens* ATCC 274 and Zorya II from *E. coli* ATCC 8739 provide protection against several phages from the Durham collection[35]. Zorya I confers protection against φAlma and φMav and deletion of each component abolishes this phenotype (Fig. 2a and Supplementary Fig. 3a). Zorya II elicits anti-phage activity against φT7 and φCS16F (Fig. 2b and Supplementary Fig. 3b). For φT7, deletion of each component leads to loss of protection (Fig. 2b). For both Zorya I and Zorya II, the peptidoglycan-binding domain of ZorB and the cytoplasmic extension of ZorA are required for defense against all tested phages (Fig. 2a, b and Supplementary Fig. 3a, b).

Mutational analysis additionally highlighted that for Zorya I and Zorya II systems, the D24 residue of ZorB, implicated in ion conductance in homologous systems, is crucial for defense (Fig. 2a, b).

Residues equivalent to ZorB I T187, R203, and R254 in *E. coli* Pal lipoprotein have been previously shown to be critical for peptidoglycan binding[36]. To investigate the capacity of ZorB I and ZorB II to bind peptidoglycan we initially isolated their putative PG-binding domain. We identified ZorB I₁₆₅₋₂₈₇ and ZorB II₁₁₅₋₂₃₅ as the most stable forms of this domain for in vitro testing. We next adapted a previously used PG-pulldown assay[37] to include additional wash steps designed to reduce

**Table 1 | Cryo-EM data collection, refinement, and validation statistics**

| | *Shewanella* sp. ANA-3 type I ZorAB (EMD-43563) (PDB 8VVN) | *S. kujiense* type II ZorAB (EMD-43560) (PDB 8VVI) |
|---|---|---|
| **Data collection and processing** | | |
| Magnification | 165,000 | 165,000 |
| Voltage (kV) | 300 | 300 |
| Electron exposure ($e^-/Å^2$) | 54.0 | 57.0 |
| Defocus range (μm) | −2.0 to −0.5 | −2.0 to −0.5 |
| Pixel size (Å) | 0.723 | 0.723 |
| Symmetry imposed | C1 | C1 |
| Initial particle images (no.) | 3,905,997 | 2,729,107 |
| Final particle images (no.) | 587,313 | 366,883 |
| Map resolution (Å) | 2.2, 2.4 | 2.8 |
| FSC threshold | 0.143 | 0.143 |
| Map resolution range (Å) | 2.2–5.4 2.4–3.4 | 2.7–9.7 |
| **Refinement** | | |
| Initial model used (PDB code) | None | None |
| Model resolution (Å) | 2.4 | 2.9 |
| FSC threshold | 0.5 | 0.5 |
| Map sharpening *B* factor ($Å^2$) | −55.3, −70.3 | −85.7 |
| Model composition | | |
| Non-hydrogen atoms | 12938 | 7762 |
| Protein residues | 1610 | 968 |
| Ligands | Na: 2 | – |
| *B* factors ($Å^2$) | | |
| Protein | 55.0 | 43.2 |
| Ligand | 22.0 | – |
| R.m.s. deviations | | |
| Bond lengths (Å) | 0.004 | 0.002 |
| Bond angles (°) | 0.600 | 0.445 |
| Validation | | |
| MolProbity score | 1.71 | 1.66 |
| Clashscore | 9.89 | 7.21 |
| Poor rotamers (%) | 0.00 | 0.00 |
| Ramachandran plot | | |
| Favored (%) | 96.9 | 96.1 |
| Allowed (%) | 3.1 | 3.9 |
| Disallowed (%) | 0 | 0 |
| CC (mask) | 0.82 | 0.80 |

nonspecific protein retention, enhancing the specificity of the protocol (see Methods). With this approach we confirmed ZorB I$_{165-287}$ and ZorB II$_{115-235}$ can bind to peptidoglycan in vitro (Fig. 2e, f) and this binding is disrupted by mutations of H186, L199, R203, R254 and R259 in ZorB I$_{165-287}$ and of H141, S143, and R215 and R230 in ZorB II$_{115-235}$ within the predicted peptidoglycan binding (Fig. 2g, h).

We then analyzed these mutations in the context of the complete assembly in vivo. In ZorB I, mutation of conserved residues proximal to the putative PG-binding site (H186, L199, R203, R254, and R259) causes loss of anti-phage activity (Fig. 2i, Supplementary Fig. 3c, and Supplementary Data 4). Similarly, In ZorB II, mutation of H141, S143, R215, and R230 (corresponding to H186, D188, and R254 in ZorB I) also

causes loss of anti-phage activity (Fig. 2j, Supplementary Fig. 3d and Supplementary Data 4). Taken together these in vitro and in vivo assays demonstrate that PG-binding is required for anti-phage activity.

We also tested the importance of the unresolved cytoplasmic domain by the introduction of mutations at several positions in both ZorA I and ZorA II. These all caused a loss of anti-phage activity (Fig. 2i, j and Supplementary Fig. 3c, d). For this reason, we inspected the cryoEM data and realized that, at the level of individual particles, a rod-like extension on the cytoplasmic face of ZorAB for both type I and II systems was evident (Fig. 3a, b). Re-centering the particles and/or focusing alignments on this extension generated 2D classes in which the rod could be seen emerging from the base of the ZorAB complex for both type I and II complexes. For the ZorAB type II complex further processing yielded a low-resolution volume that accounts for the predicted first helical domain of the rod (residues 110–140; 11% of total rod residues) and allows docking of AlphaFold models (Fig. 3c) for this region to the high-resolution complexes (Fig. 3d). The volumes support modeling of this region as a 5-helix bundle and suggest that it emerges at an angle to the vertical axis of the complex (Fig. 3d, e). Given that this bundle is attached to the presumed rotating ZorA components we hypothesize that ion flow through the complex and rotation of ZorA either leads to rearrangement of the bundle or drives rotation of the rod through the cytoplasm.

### Zorya I and II mediate population-wide immunity

Previous studies have reported varying behaviors of Zorya systems. Some have indicated that Zorya systems operate through conditional abortive infection[2], whereas more recent research suggests that Zorya I functions through a direct defense mechanism[37,38]. To investigate this further, we followed the dynamics of Zorya I and Zorya II-mediated defense over a 12-h infection period.

At high MOI (MOI = 5), cells over-expressing Zorya I exhibited reduced φAlma titers over time relative to the vector control strain and a culture collapse (Fig. 4a–c and Supplementary Fig. 4a–c). At lower MOI (MOI = 0.05), the number of phages released remains lower compared to vector control, while the growth rate and CFU counts of Zorya I cells increase (Fig. 4d–f and Supplementary Fig. 4d–f), consistent with a population-wide immunity mechanism.

For Zorya II, over-expression of the system during infection with φT7 at MOI = 5 leads to rapid loss of cell viability (Fig. 4g–i and Supplementary Fig. 4g–i). Nevertheless, the φT7 titer remains unchanged over time in Zorya II cells, indicating that Zorya II reduces bacterial fitness to prevent phage propagation (Fig. 4g and Supplementary data Fig. 3g–i). Consistent with a population-wide immunity phenotype, when infected with low φT7 MOI (MOI = 0.05), Zorya II-cells exhibit an unaltered growth rate and inhibition of phage propagation (Fig. 4j–l and Supplementary data Fig. 3j–l).

To confirm that the observed phenotypes were not artifacts of protein overexpression, we conducted the same experiments by expressing Zorya I and II from their native promoters.

Under these conditions, Zorya I did not confer protection against φMav or φAlma, while Zorya II activity was markedly reduced against φT7 and φCS16F (Fig. 5a). Although additional phages susceptible to Zorya I were not identified, Zorya II, expressed from its native promoter, strongly inhibited several newly isolated phages, including φphAvM, φphGM01, and φEsilda (Fig. 5a). Importantly, protection against φphAvM was completely abolished when any component of Zorya II was deleted, even under native expression conditions (Fig. 5b).

To further investigate the dynamics of Zorya II-mediated defense under native conditions, we focused on φphAvM due to its robust inhibition. At high φphAvM MOI (MOI = 5), Zorya II-cells, under native expression conditions, maintain a static $OD_{600nm}$, showing neither growth nor lysis, indicative of a bacteriostatic phenotype. In contrast,

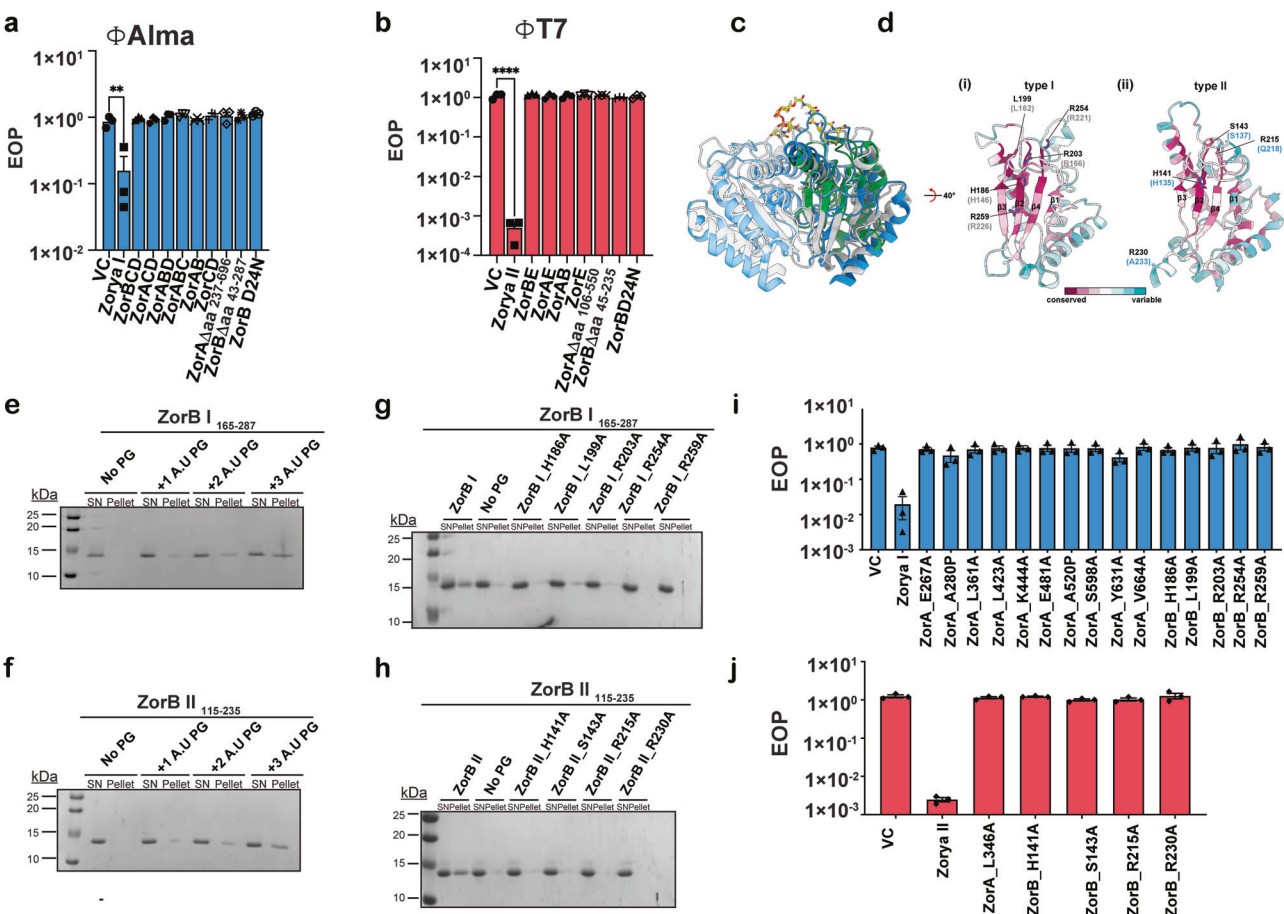

**Fig. 2 | ZorA and ZorB unique structural features are central for phage defense.**
**a** Efficiency of plating (EOP) measurement for *E. coli* MT56 carrying empty vector (VC, pGM39, a pT12 backbone vector with no insert[32]) or the same plasmid encoding Zorya I, ZorA I, ZorB I, ZorABC, ZorABD, ZorBDC, ZorAB I, ZorCD, ZorB D24N, ZorA IΔaa_{237-696}, and ZorB IΔaa_{43-287} when infected with ϕAlma. Points show mean ± SEM (*n* = 3 biological replicates). Statistical significance for each panel was calculated with Graphpad applying a one-way ANOVA with Dunnett's multiple comparison test. No significance was detected, unless indicated (*$p \leq 0.05$). For the VC vs Zorya I comparison, *p*-value = 0.0028. **b** Efficiency of plating (EOP) measurement for *E. coli* MT56 carrying the empty vector (VC, pGM39) or the same plasmid encoding Zorya II, ZorA II, ZorB II, ZorAB II, ZorE, ZorB D24N, ZorA IΔaa_{106-550} and ZorB IΔaa_{45-235} when infected with phage ϕT7. Points show mean ± SEM (*n* = 3 biological replicates). Points show mean ± SEM (*n* = 3 biological replicates). Statistical significance for each panel was calculated with Graphpad applying a one-way ANOVA with Dunnett's multiple comparison test. No significance was detected, unless indicated (*$p \leq 0.05$). For VC vs Zorya II comparison, *p*-value = 0.00001852. Induction of each construct in panels (**a**, **b**) was performed by the addition of 0.02% L-Rhamnose. **c** Overlay of the ZorB PG-binding domain of type I ZorA_5B_2 from *Shewanella* sp. strain ANA-3 (gray) with type II ZorA_5B_2 from *Sulfuricurvum kujiense* (blue) and *H. influenza* peptidoglycan-associated lipoprotein (Pal; green) with bound peptidoglycan in yellow (PDB 2AIZ). Cα RMSD between Shewanella sp. strain ANA-3 ZorB PG-binding domain and 2AIZ is 2.43 Å; Cα RMSD between Sulfuricurvum kujiense ZorB PG-binding domain and 2AIZ is 2.78 Å; Cα RMSD between both species ZorB PG-binding domains is 1.93 Å. **d** Conserved residues cluster proximal to the proposed PG-binding site in *Shewanella* sp. strain

ANA-3 ZorB (i) and *S. kujiense* ZorB (ii). Models are colored based on conservation scores; indicated residues diminished phage infection when mutated to alanine. Black residue labels are numbered according to the corresponding residues for *Serratia marcescens* ATCC 274 (type I) or *E. coli* ATCC 8739 (type II) used in the infection assays; gray and blue residue numbers correspond to the numbering of the modeled *Shewanella* and *Sulfuricurvum* complexes, respectively. **e**, **f** Peptidoglycan pull-down assay when incubated with purified ZorB I_{165–287} (**e**) and ZorB II_{115–235} (**f**). **g**, **h** Peptidoglycan pull-down assay when incubated with purified ZorB I_{165–287} and its mutants as indicated in panel (**g**), or of and ZorB II_{115–235} and its mutants as shown in panel (**h**). **i** Efficiency of plating (EOP) measurement for *E. coli* MT56 harboring empty vector (VC, pGM39), the same plasmid encoding wild-type Zorya I or a version of Zorya I where single point mutations were introduced in ZorA I or ZorB I, as indicated in panel (**j**) when infected with ϕAlma. The full set of mutations tested is reported in Supplementary Fig. 3c. For panels **e**–**h** images are representative of three independent experiments. **j** Efficiency of plating (EOP) measurement for *E. coli* MT56 expressing empty vector (VC, pGM39), the same plasmid encoding wild-type Zorya II or a version of Zorya II carrying point mutations in ZorA II or ZorB II, as indicated in panel (**i**), when infected with ϕT7. The full set of mutations tested is reported in Supplementary Fig. 3d. Points show mean ± SEM (n = 3 biological replicates). Induction of each construct in panels (**i**, **j**) was performed by the addition of 0.02% L-Rhamnose. Statistical significance for each panel was calculated with Graphpad applying a one-way ANOVA with Dunnett's multiple comparison test. No significance was detected, unless indicated (*$p \leq 0.05$). For panels **i**, **j** the statistical analysis results are reported in Supplementary Data 4.

CFU counts decline slightly relative to *t* = 0 h (Fig. 5c–e and Supplementary Fig. 5). Importantly, the ϕphAvM titer remains unchanged over the 12-h period, ruling out phage evasion as the cause of decreased bacterial fitness. At low ϕphAvM MOI (MOI = 0.05), Zorya II-cells exhibit normal growth and effectively inhibit phage propagation (Fig. 5f–h and Supplementary Fig. 5). Together, these findings demonstrate that, even under native expression conditions, Zorya II-

mediated defense enforces population-wide immunity, restricting phage proliferation at the cost of bacterial fitness.

## ZorE mode of action includes single-stranded DNA breaks
Prior work reported that ZorE carries an HNH endonuclease domain whilst ZorD carries a Mrr-like nuclease domain[2] (Supplementary Fig. 1b, c).

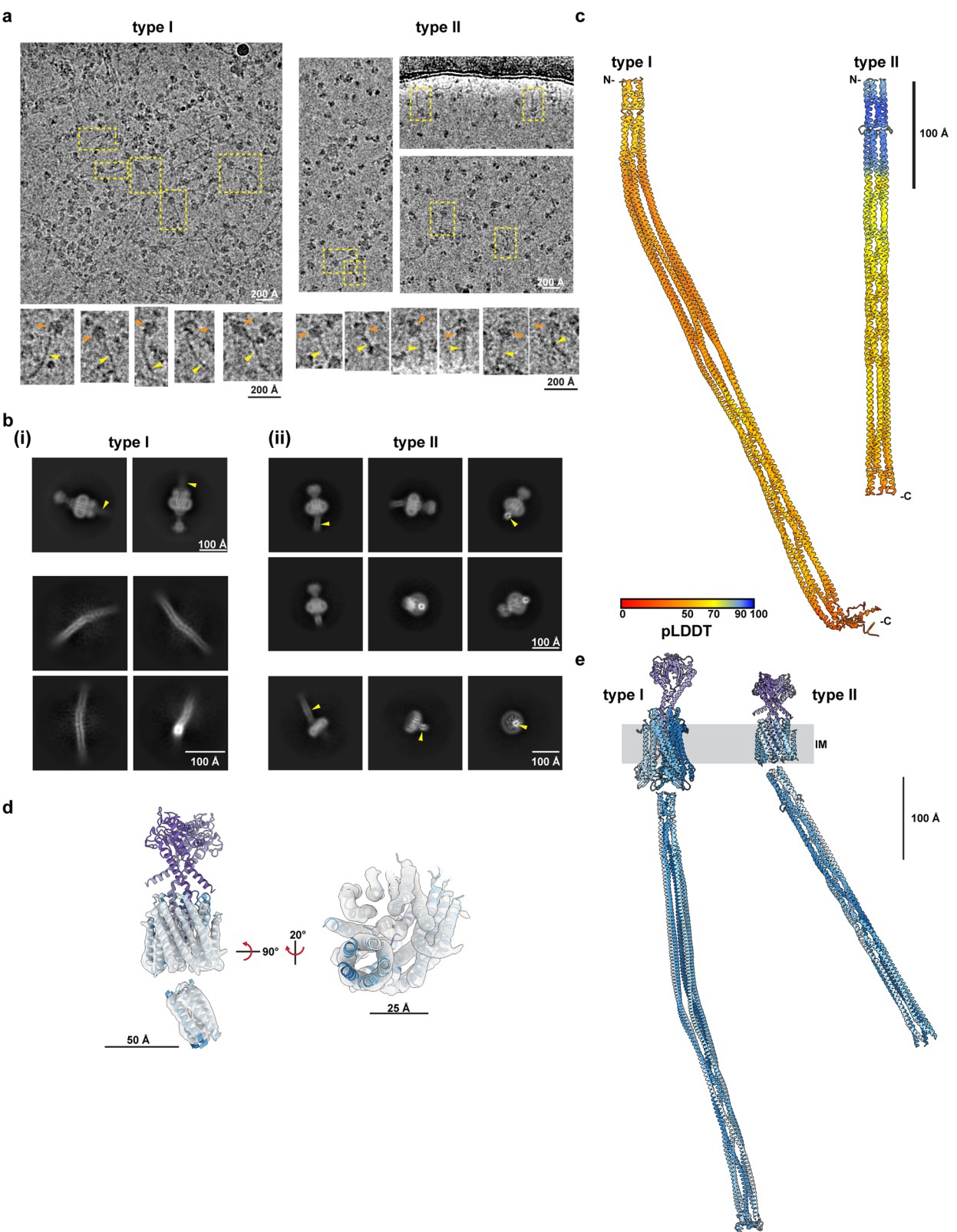

To characterize ZorE activity, we fused a Twin Strep-tag to its C-terminus and confirmed the functionality of this tagged version in vivo (Supplementary Fig. 6a). We determined that purified ZorE-Strep is monomeric in solution using Mass Photometry(Refeyn) and Size Exclusion Chromatography (Supplementary Fig. 6b–d). To assess whether ZorE selectively degrades specific DNA sources, we compared its ability to target phage and bacterial genomic DNAs. ZorE degrades both *E. coli* chromosomal DNA and phage genomic DNA, but only at high protein concentrations, with phage DNA exhibiting greater susceptibility (Supplementary Fig. 6e–h).

Next, we utilized plasmid pSG483 to examine specific nuclease activities of ZorE. Titration of purified ZorE against a constant concentration of supercoiled pSG483 DNA revealed that ZorE prevalently exhibits nicking activity, converting the supercoiled form of pSG483

**Fig. 3 | Cytoplasmic residues of ZorA form an extended pentameric rod. a** Eight angstrom lowpass-filtered representative cryo-EM micrographs of purified ZorAB complexes (top) from *Shewanella sp.* strain ANA-3 (left) or *Sulfuricurvum kujiense* (right). Enlarged selected particles are depicted below the micrographs with their corresponding micrograph locations outlined as yellow boxes. Yellow arrowheads denote the cytoplasmic rods, and orange arrowheads denote the core complex embedded within the detergent micelle. Similar micrographs were obtained from three independent preparations of each sample. **b** (i) Two-dimensional averages of cryo-EM ZorAB particles from *Shewanella sp.* strain ANA-3 from focused classifications of the core (top) or rod structure (bottom). (ii) 2D averages of cryo-EM ZorAB particles from *S. kujiense* from focused classifications of the core (top) or

partial core after particle subtraction (bottom). **c** Pentameric models of the ZorA cytoplasmic extensions based on Alphafold multimer predictions (residues 237–696 of *Shewanella*; left, residues 110–378 of *Sulfuricurvum*; right). Models colored by pLDDT score. **d**, Docking residues 111–140 of the *S. kujiense* ZorA cytoplasmic rod Alphafold model (residues 111–140) into a ~4.9 Å ZorAB cryo-EM volume containing partial helical rod density, with views shown from the side (left) or from the cytoplasm (right). **e** Full ZorAB models from type I (*Shewanella sp.* strain ANA-3; left) and type II (*Sulfuricurvum kujiense*; right) based on cryo-EM models of the ZorA₅B₂ core and Alphafold models of the cytoplasmic ZorA rod. Models in panels (**e**, **f**) were colored as in Fig. 1a, b.

into the nicked form (Fig. 6a, b). Additionally, at higher concentrations of ZorE, we observed the formation of a small percentage of linear DNA (~20%), suggesting limited double-strand cleavage at elevated protein levels(Fig. 6a, b).

To further characterize ZorE's activity, we tested the influence of various metal cations. We observed variable effects across the tested cations, with $Mg^{2+}$ and $Mn^{2+}$ producing the most significant enhancement of nicking activity (Supplementary Fig. 6i, j). Notably, $Mn^{2+}$ had a modest yet more pronounced effect compared to other ions in promoting the formation of linear pSG483 DNA, suggesting its role in facilitating limited double-strand breaks(Supplementary Fig. 6i, j).

Finally, we conducted a time-course experiment with a constant concentration of ZorE tested against various DNA forms: supercoiled, relaxed, linear, and nicked. We found that ZorE had no effect on linear DNA (Supplementary Fig. 6k, l). At later time points, ZorE converted the nicked form to linear DNA, albeit with very low efficiency (Supplementary Fig. 6m, n). Relaxed and supercoiled DNA were primarily converted to the nicked form; however, ZorE processed relaxed DNA less efficiently than supercoiled DNA and did not generate any linear DNA from relaxed DNA (Fig. 6c–f).

These results indicate that ZorE's main activity is nicking, though at elevated concentrations, it can also induce double-strand breaks or generate nicks on opposing strands at later stages.

## Population-wide immunity is achieved through host chromosomal DNA damage

Previously it has been proposed that Zorya may act to depolarize host cells as a mechanism of abortive infection[2]. However, using the voltage-sensitive dye $DiBAC_4(3)$, which accumulates in depolarized cells and emits green fluorescence, we observed no indication of Zorya-mediated depolarization (Supplementary Fig. 7a). To exclude that depolarization only occurs in response to a phage trigger, we measured the fluorescence of $DiBAC_4(3)$ and propidium iodide (which enters bacterial cells with impaired membrane integrity, causing red fluorescence), over the course of phage infection. Only cells harboring an empty vector show increased green and red fluorescence, indicative of phage-mediated killing (Supplementary Fig. 7b, c). Additionally, monitoring resazurin fluorescence revealed no detectable alteration in metabolic activity upon Zorya I and Zorya II expression (Supplementary Fig. 7d).

We next investigated whether Zorya decreases the fitness of infected cells through damage to the bacterial chromosome. We observed that cells expressing the full Zorya II system or ZorE alone (but not ZorAB II) appear elongated and exhibit regions where 4′,6-diamidino-2-phenylindole (DAPI) fluorescence was reduced or absent (Fig. 6g and Supplementary Fig. 7e), consistent with DNA damage. The presence of DNA damage in ZorE-only cells is likely due to its over-expression. In the case of Zorya I expression, alterations of the DAPI signal and prominent cell elongation are not detectable (Fig. 6g and Supplementary Fig. 7e).

To confirm whether genomic DNA lesions were present following the expression of Zorya components, we adapted a previously published electrophoresis-based assay (EAsy-GeL)[39]. Here chromosomal

DNA is isolated in a neutral buffer, allowing detection of double-stranded breaks (DSBs). Subsequent incubation in alkaline conditions further reveals alkaline unwinding-sensitive sites (AU-SSs), mainly caused by single-strand breaks (SSBs)[39]. Altered migration of DNA in a neutral buffer (detected as smaller bands and/or a smear) is indicative of the prevalence of DSBs, whereas altered migration of DNA in an alkaline buffer indicates the presence of SSBs.

DNA extracted from cells over-expressing either the full Zorya I system or only ZorCD exhibit significant changes in migration after alkaline treatment, indicating that Zorya I effector proteins likely induce single-strand breaks (SSBs) in the host chromosome (Fig. 6h). A similar pattern was also observed during φAlma infection (Supplementary Fig. 7f).

Conversely, DNA purified from cells over-expressing Zorya II and ZorE showed a smeary appearance, both in the presence or absence of φT7 infection (Fig. 6i and Supplementary Fig. 7g). This phenotype is further exacerbated after alkaline treatment (Fig. 6i and Supplementary Fig. 7g), demonstrating that Zorya II, through ZorE, can damage the host DNA by introducing SSBs and, to a lesser extent, DSBs, consistent with what was observed in vitro (Fig. 6a–f). Notably, under native expression conditions and in the absence of phage infection, the DNA of Zorya II cells remains intact (Fig. 6j). In the presence of φphAvM, Zorya II-cells exhibit a pattern similar to that seen under over-expression conditions, indicating that ZorE retains its ability to preferentially introduce single-strand breaks (SSBs) in both bacterial and phage DNA under native conditions (Fig. 6j). We note that under native conditions, ZorE-only cells do not exhibit detectable DNA-targeting activity, highlighting that they require ZorAB for recruitment or activation (Fig. 6j).

Given that ZorE induced degradation of host or chromosomal DNA only at higher concentrations in vitro and that ZorE-only mediated DNA targeting was absent under native conditions in vivo without ZorAB II, we sought to investigate its recruitment dynamics in vivo through a cellular fractionation experiment. Our findings reveal that, under native expression conditions, the ZorAB complex specifically directs ZorE to the membrane during phage infection (Fig. 6k). In contrast, in the absence of infection, ZorE predominantly localizes to the cytoplasm, suggesting that its membrane association is specifically triggered by phage infection (Fig. 6k).

We conclude that ZorAB II-mediated recruitment likely increases ZorE's local concentration, thereby enhancing its ability to efficiently target DNA during the defense process.

## Discussion

In this study we report mechanistic and structural insights into the Zorya phage defense systems, demonstrating that the core components ZorA and ZorB form a macromolecular complex, reminiscent of flagellar proteins MotA and MotB.

Beyond similarity to the MotAB core, the ZorA-ZorB complex exhibits unique features, such as distinctive and flexible, rod-like cytoplasmic extensions for ZorA. We show that these domains are crucial for Zorya I and Zorya II anti-phage activity. We also demonstrate that ZorB can bind to peptidoglycan and that mutations in the PG-binding domain

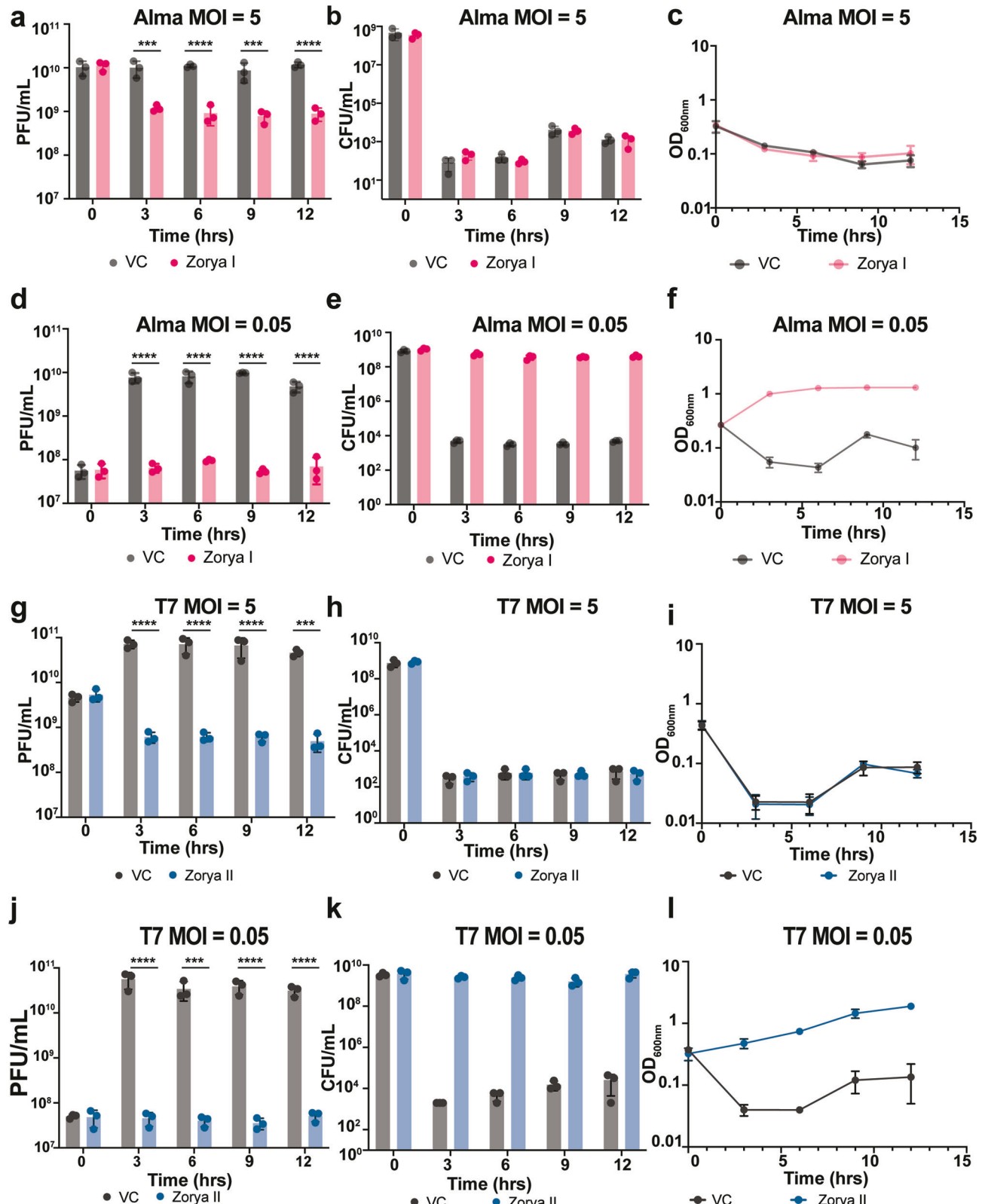

abolish peptidoglycan binding and anti-phage activity. We suggest that in the absence of a phage threat, ZorB is in a resting state, not bound to peptidoglycan. Upon phage infection, the peptidoglycan layer is either pushed down or damaged, facilitating ZorB-mediated binding. This, in turn, may trigger a conformational change in both ZorA and ZorB, leading to the opening of the ion channel likely by pulling up of the ZorB component relative to ZorA. By analogy with the MotAB system, ion

flow through the opened channel will then lead to rotation of ZorA with respect to the PG-tethered ZorB. However, we note that a limitation of this study is that direct observation of peptidoglycan binding by ZorB in vivo during phage infection is currently not feasible, presenting a challenge in fully validating this model.

Previous studies have suggested that Zorya systems operate through abortive infection[2], however, more recent reports have shown

**Fig. 4 | Zorya systems prevent phage infection through population-wide immunity. a–f** *E. coli* MT56 harboring VC or Zorya I was grown in LB supplemented with 0.02% L-Rhamnose and infected with φAlma at MOI 5 or 0.05. The (**a**, **d**) titer (PFU/mL), **b**, **e** cell counts (CFU/mL), and **c**, **f** growth rate (OD$_{600nm}$) of each culture was measured at several time points, as shown in panels (**a–f**), over the course of 12 h post-infection. **g–l** *E. coli* MT56 carrying VC or Zorya II were grown in LB supplemented with 0.02% L-Rhamnose and infected with φT7 at MOI 5 or 0.05. The **g**, **j** titer (PFU/mL), **h**, **k** cell counts (CFU/mL), and **i**, **l** the growth rate (OD$_{600nm}$) of each culture were measured at several time points, as shown in panels (**g–l**), over the course of 12 h post-infection. For all panels, points show mean ± SEM (*n* = 3 biological replicates). Statistical significance was calculated with Graphpad applying a two-way ANOVA comparison test. No significance was detected, unless indicated ($p \leq 0.05$). In **a**, the *p*-values for the VC vs Zorya I comparisons are as follows: for the 3-h time point, $p = 0.0003$; for the 6-h time point, $p < 0.0001$; for the 9-h time point, $p = 0.0009$; and for the 12-h time point, $p < 0.0001$. **d** The *p*-values for the VC vs Zorya I comparisons are as follows: for the 3-h time point, $p < 0.0001$; for the 6-h time point, $p < 0.0001$; for the 9-h time point, $p = 0.0009$; and for the 12-h time point, $p < 0.0001$. **g** The *p*-values for the VC vs Zorya II comparisons are as follows: for the 3-h time point, $p < 0.0001$; for the 6-h time point, $p < 0.0001$; for the 9-h time point, $p = 0.0009$; and for the 12-h time point, $p < 0.0001$. **j** The *p*-values for the VC vs Zorya II comparisons are as follows: for the 3-h time point, $p < 0.0001$; for the 6-h time point, $p = 0.0002$; for the 9-h time point, $p = 0.0009$; and for the 12-h time point, $p = 0.0004$.

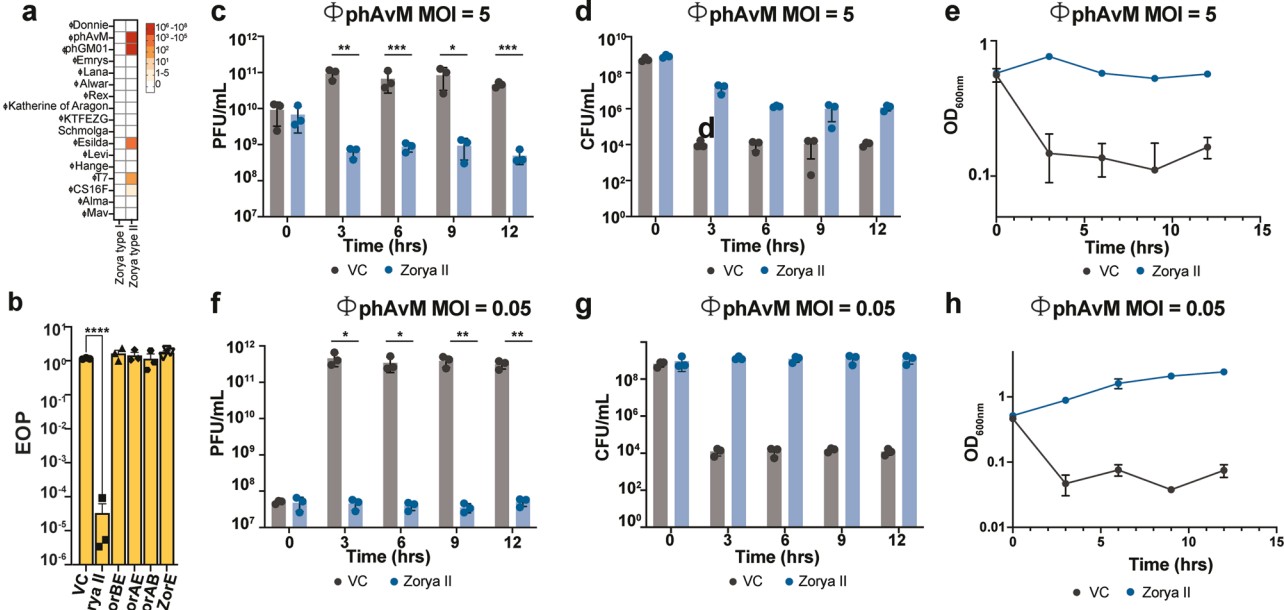

**Fig. 5 | Natively-expressed Zorya II halts phage infection by mediating population-wide immunity. a** The anti-phage activity of Zorya I and Zorya II under the control of their native promoter was evaluated by calculation of their fold protection against a suite of newly isolated environmental coliphages. As a control, φAlma, φMav, φT7, and φCS16F were included. Fold protection was calculated by dividing the value of efficiency of plating (EOP) for strains expressing Zorya I or Zorya II by the EOP value of a strain carrying the empty vector (pSUPROM), when infected with phages as shown in panel (**a**). **b** Efficiency of plating (EOP) measurement for *E. coli* MT56 carrying the empty vector (VC, pSUPROM) or the same vector carrying Zorya II, ZorAE, ZorBE, Zor AB II or ZorE under the control of their native promoter, when infected with φphAvM. Points show mean ± SEM (*n* = 3 biological replicates). Statistical significance was calculated with Graphpad applying a one-way ANOVA with Dunnett's multiple comparison test. No significance was detected, unless indicated ($p \leq 0.05$). For VC vs Zorya II comparison, *p*-value = <0.0001. **c–h** *E. coli* MT56 carrying the empty vector (VC, pSUPROM) or the same vector expressing Zorya II under its native promoter was infected with φphAvM at MOI 5 or 0.05. The **c**, **f**, titer (PFU/mL), **d**, **g** cell counts (CFU/mL) and **e**, **h**, and **g**, the growth rate (OD$_{600nm}$) of each culture was measured at several time points, as shown in panels (**c–h**), over the course of 12 h post-infection. For panels **c–h**, points show mean ± SEM (*n* = 3 biological replicates). Statistical significance was calculated with Graphpad applying a two-way ANOVA comparison test. No significance was detected, unless indicated ($p \leq 0.05$). **c** The *p*-values for the VC vs Zorya II comparisons are as follows: for the 3-h time point, $p = 0.0066$; for the 6-h time point, $p = 0.045$; for the 9-h time point, $p = 0.049$; and for the 12-h time point, $p = 0.0006$. **f** The *p*-values for the VC vs Zorya II comparisons are as follows: for the 3-h time point, $p = 0.013$; for the 6-h time point, $p = 0.019$; for the 9-h time point, $p = 0.007$; and for the 12-h time point, $p = 0.0028$.

instances of Zorya I acting as a first-line defense[37,38]. Our data show that during Zorya I and Zorya II-mediated defense, impairment of the fitness of infected cells is part of the mechanism (Figs. 4 and 5). Importantly, previous studies tested Zorya systems in *E. coli* K12, whereas our experiments utilized a B strain, suggesting that the genetic background may contribute to the observed discrepancies[37,38].

When investigating the mechanism by which Zorya-mediated defense decreases bacterial fitness, we found that Zorya II induces host chromosome damage and cell elongation (Fig. 6 and Supplementary Fig. 6). Both in vivo and in vitro, ZorE predominantly exhibits nicking activity (SSBs) and introduces some DSBs at higher concentrations or after longer incubation times. The lower frequency of DSBs observed may result from either the increased ZorE concentration or additional nicks on the complementary strand

(Fig. 6). Furthermore, efficient degradation of phage and chromosomal DNA is only observed at high concentrations of ZorE in vitro. As localization data suggest that ZorAB mediates ZorE's recruitment specifically at the site of infection and exclusively following a phage attack, it is likely that this process is responsible for increasing ZorE's local concentration and facilitating its optimal nickase activity against chromosomal and phage DNA during infection.

DNA nickases drive population-wide immunity in several other anti-phage systems, such as Hma and Gabija[40–43]. Notably, negative DNA supercoiling is correlated with bacterial growth phases, and its imbalance—such as that caused by the introduction of single-strand breaks—has been shown to disrupt key gene expression[44,45]. These alterations caused by SSBs and loss of DNA supercoiling, could account for the growth defect caused by ZorE.

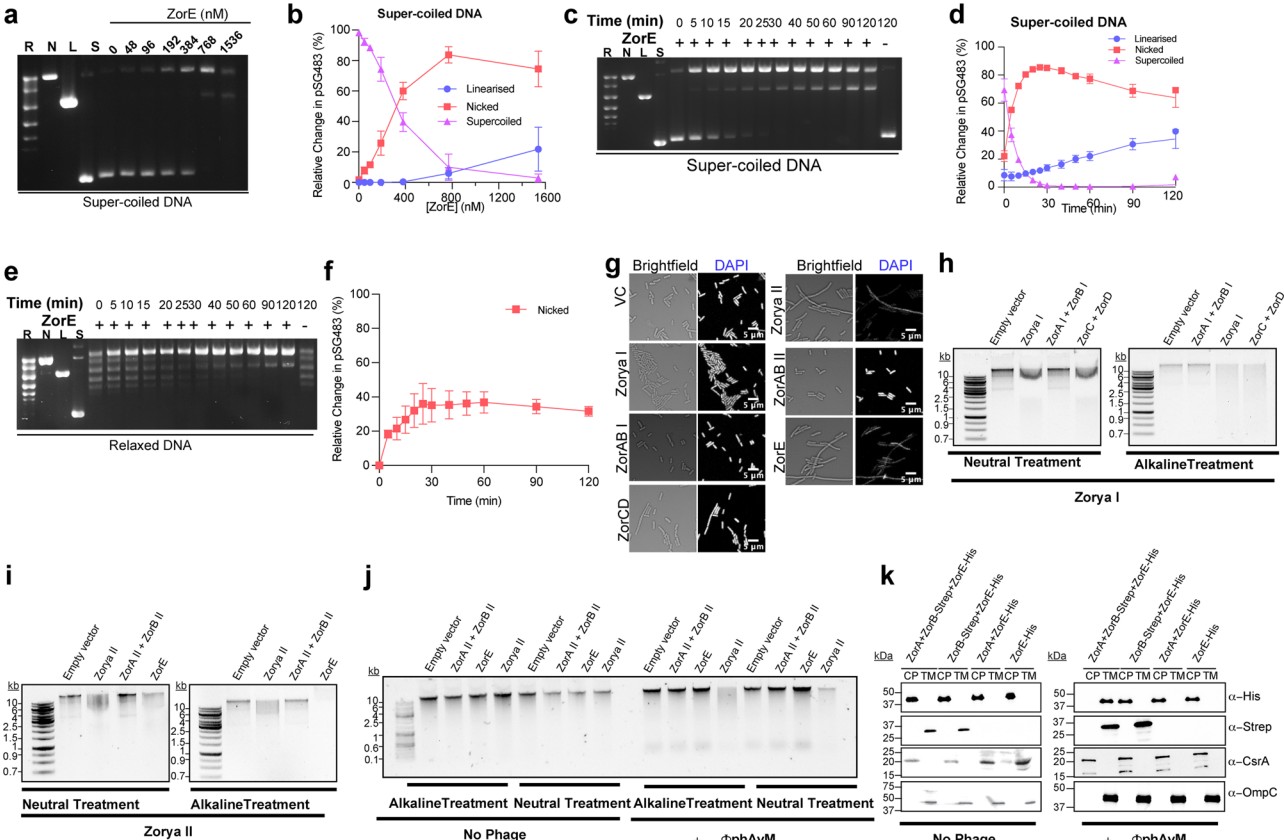

**Fig. 6 | ZorE can degrade phage and chromosomal DNA. a** ZorE-Strep titrated against a constant amount of supercoiled pSG483 plasmid DNA (6 nM). Samples were incubated at 37 °C for 60 min in the presence of 5 mM Mg $^{2+}$. **b** Densitometry quantification of nicking of pSG483 by ZorE as shown in panel (**a**). **c** ZorE (768 nM) was incubated with supercoiled plasmid pSG483 (6 nM) at 37 °C for 0 to 60 min with 5 mM Mg$^{2+}$. **d** Densitometry quantification of nicking of pSG483 by ZorE as shown in panel (**c**). **e** ZorE (768 nM) was incubated with relaxed plasmid pSG483 (6 nM) for 0 to 60 min with 5 mM Mg$^{2+}$ at 37 °C. **f** Densitometry quantification of nicking of pSG483 by ZorE as shown in panel (**d**). For panels **a**, **c**, and **e**, reactions were stopped by the addition of EDTA and SDS, and products were analyzed by gel electrophoresis in a 1× TAE, 1.4% agarose gel, post-stained with ethidium bromide. In all gels, control lanes represent forms of plasmid pSG483; R, relaxed (multiple topoisomers); N, nicked; L, linear; S, supercoiled. For panels **b**, **d**, and **f** densitometry was performed using ImageJ (version 1.54g) with background subtracted and band intensity measured in triplicate. The percentage of nicked, linear, and supercoiled pSG483 DNA of the total pSG483 DNA per lane was determined by calculating the average intensity (*n* = 3) of each lane's nicked, linear, and supercoiled bands, respectively. as a percentage of the total average intensity of all bands per lane. Relative band intensity was determined by normalizing the average (*n* = 3) intensity of the "0 µM ZorE" lane to 100% and taking the average intensity of the subsequent lanes' bands as a percentage of the "0 µM ZorE" lane. Error bars represent the standard error of the mean of triplicate data. **g** *E. coli* MT56 harboring empty vector (VC, pGM39) or the same plasmid encoding Zorya I, ZorAB I, ZorCD,

Zorya II, ZorAB II, or ZorE were grown in LB supplemented with 0.2% L-Rhamnose for 2 h. Following incubation, cells were stained with DAPI and imaged by fluorescence microscopy. Scale bar 5 µm. **h** *E. coli* MT56 harboring empty vector (VC, pGM39) or the same plasmid encoding Zorya I, ZorAB I, or ZorCD were grown as in (**g**) and total genomic DNA (gDNA) was extracted. Neutral and alkaline treatment of gDNA, followed by electrophoretic analysis (Methods) was used to assess for DNA breaks. **i** *E. coli* MT56 harboring empty vector (VC, pGM39) or plasmids expressing Zorya II, ZorAB II or ZorE were induced as in (**g**) and total gDNA was isolated as in (**h**). Genomic DNA was subjected to neutral and alkaline treatment, as described in "Methods", and subjected to electrophoretic analysis. **j** *E. coli* MT56 carrying the empty vector (VC, pSUPROM) or the same plasmid harboring Zorya II, ZorAB II, or ZorE under the control of their native promoter were grown in the presence and absence of ϕphAvM (MOI 0.1) until first burst event. Total gDNA was extracted and subjected to neutral and alkaline treatment as in panels (**h–i**). **k** *E. coli* MT56 carrying the empty vector (VC, pSUPROM) or the same plasmid harboring Zorya II, ZorAB II, or ZorE under the control of their native promoter were grown in the presence and absence of ϕphAvM (MOI 0.1). Strains were fractionated to produce a soluble cytoplasmic fraction (CP) and a total membrane fraction (TM). Samples were analyzed by immunoblot with antibodies to the His$_6$ tag (for detection of ZorE) and Strep-tag (for detection of ZorB). OmpC was used as a membrane control and CsrA as cytoplasmic control. For panels **g–k**, gels are representative of three independent experiments.

Whilst we were not able to detect any significant proof of DNA damage in fluorescence microscopy, EASy-GeL assays suggest that host DNA damage also occurs for Zorya I (Fig. 6). ZorAB-mediated recruitment of ZorCD and the ability of ZorD to target DNA has been reported[37]. This suggests a conserved defense mechanism across all subtypes, involving ZorAB-mediated sensing and recruitment of a nuclease-like complex[31].

Based on these findings, we suggest a model (Fig. 7) by which ZorA and ZorB assemble to form a rotary motor that can sense the incoming phage threat. Such detection is dependent on ZorB PG-binding, which may act via sensing damage to the peptidoglycan layer. Our data and previous work show that Zorya systems require an intact ion flux

through ZorAB to be functional (Fig. 7)[2]. This flux could translate the 'threat' signal through the cell. The flexibility of the ZorA rod-like structures may suggest that signal translation via rotation of ZorA with respect to ZorB causes a conformational change of the ZorA cytoplasmic regions, which recruits a nuclease complex such as ZorCD and ZorE increasing the local concentration of nuclease at the site of phage attack to the levels needed for significant activity (Fig. 7).

Interestingly the Zorya systems represent a unique example where part of a conserved bacterial macromolecular machinery, such as the flagellar motor, has been adapted to provide a complex and modular defense strategy against mobile genetic elements (MGEs). As efforts towards the discovery and characterization of anti-phage

**(i) No Infection**

**(ii) Phage infection**

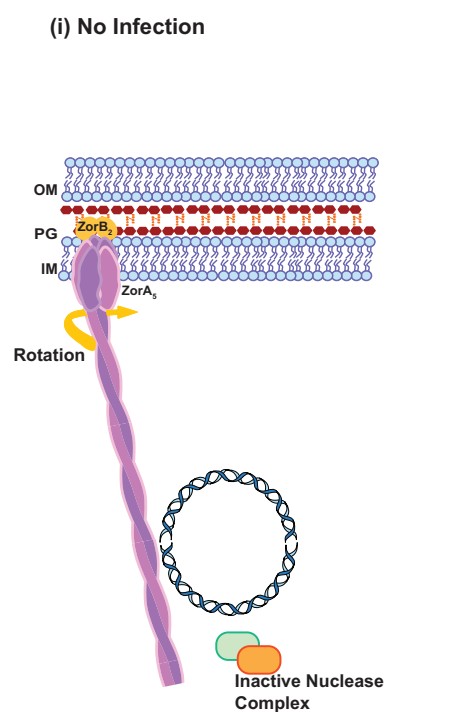
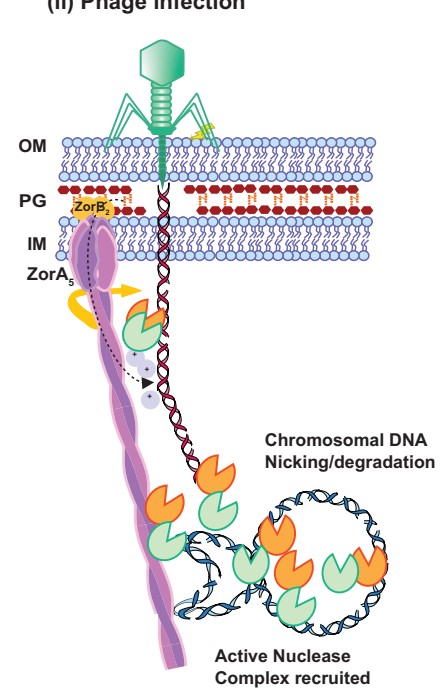

**Fig. 7 | Proposed model for Zorya anti-phage mechanism.** (i) Inactive Zorya systems are embedded within the bacterial inner membrane. (ii) Upon infection, phage-mediated puncturing pushes down the peptidoglycan (PG) layer or causes PG damage, an event that can be 'sensed' by the ZorB PG-binding domain. This triggers a conformational change in ZorAB, opening the ion channel and leading to the rotation of ZorA rod-like extensions within the cytoplasm. ZorA-rod will recruit a nuclease complex to prevent phage infection by predominantly targeting bacterial chromosomes. In the case of Zorya II, ZorAB II-mediated recruitments increase the local concentration of ZorE, enabling full efficiency of its nicking activity. Created in BioRender. Mariano, G. (2025) https://BioRender.com/w85a533.

strategies increase, it is tempting to speculate that more examples of the adaptation of bacterial macromolecular machinery for anti-phage defense may have occurred.

## Methods
### Bacterial strains, plasmids, and culture conditions
The strain *E. coli* MT56, a derivative of *E. coli* BL21-DE3[46] optimized for membrane protein expression, was grown at 37 °C on solid media or in liquid culture. For liquid growth, Luria broth (LB) was used as the standard medium, and cultures were incubated shaking at 200 rpm. For growth on solid media, LB was supplemented with 1.5% (w/v) agar for agar plates and with 0.35% (w/v) agar for soft agar used for top-lawns. When required, LB was supplemented with Kanamycin (Kan, 50 µg/mL) or ʟ-Rhamnose (0.2% or 0.02% w/v), as detailed below. Expression of Zorya systems was performed from a pT12-derived plasmid, with a ʟ-Rhamnose-induced promoter. A MotAB-Strep insert was deleted from the vector using a KLD enzyme mix (See below) to obtain the empty vector pGM39[32]. All Zorya systems and derivatives were then cloned in pGM39. Strains and plasmids used in this study are listed in Supplementary Table 1 and primers and cloning strategies are listed in Supplementary Table 2.

### DNA manipulation and transformation
Plasmid backbones and inserts for cloning were amplified using Q5 High-Fidelity DNA Polymerase (NEB). PCR products and plasmids were purified with Monarch DNA kits (NEB). Cloning of inserts was performed with NEBuilder assembly and overlapping primers for amplification were designed using the NEBuilder assembly tool (https://nebuilderv1.neb.com/) (Supplementary Table 2). Plasmids and inserts were assembled using NEBuilder HiFi DNA Assembly (NEB), followed by incubation at 50 °C for 60 min. Deletions of single Zorya components and the ZorBD24N point mutations were performed with KLD enzyme mix (NEB). Point mutations in ZorA cytoplasmic domain and ZorB patch1, patch2, and PG-binding loop were designed with the NEBuilder assembly tool (https://nebuilderv1.neb.com/). A full list of primers is reported in Supplementary Table 2.

### Phage propagation, lysate preparation, and measurements of Efficiency of plating
Phage lysates were stored in phage buffer (10 mM Tris–HCl pH 7.4, 10 mM MgSO4, 0.1% gelatin) and propagated in *E. coli* DH5α. For propagation, neat lysates or their serial dilutions were added to 200 µL of *E. coli* DH5α and incubated for 5 min at room temperature. The mixture was added to 5 mL of soft agar and poured onto LB agar plates. Plates were then incubated at 37 °C overnight. Soft agar lawns containing confluent plaques were scraped off and mixed with 3 mL of phage buffer and 500 µL of chloroform. Mixtures were vortexed for 2 min and incubated for 30 min at 4 °C. Samples were then centrifuged at 4000×g for 20 min and the supernatant was collected and added to 100 µL of chloroform for storage.

For measurements of the efficiency of plating (EOP), 10 µL of neat phage lysate or a serial dilution of the lysate were added to 200 µL of an overnight culture of *E. coli* MT56 carrying the empty vector (Supplementary Table 2) or a vector encoding Zorya I, Zorya II or their derived mutants. Five mL of soft agar was added to each culture and poured onto LB agar plates supplemented with 0.02% ʟ-Rhamnose and kanamycin. As a control strain, plasmid-free *E. coli* MT56 was used. EOP was measured as the number of PFU/mL$^{-1}$ of a test strain divided by the number of PFU/mL of the control strain.

For the calculation of fold protection, the ratio between the EOP values on strains carrying tested Zorya homologs and the EOP value on a strain carrying an empty vector was calculated.

## Measurements of PFU/mL$^{-1}$, CFU/mL$^{-1}$, and OD$_{600nm}$ over the course of phage infection

*E. coli* MT56 harboring empty vector or the same vector encoding Zorya I, Zorya II, or their mutants were grown in LB supplemented with kanamycin and 0.02% *L*-Rhamnose monohydrate to an OD$_{600nm}$ of ~0.4. Strains carrying Zorya I and its mutants were infected with ϕAlma at MOI 0.5. Strains expressing Zorya II and its mutants were infected with ϕMak at an MOI of 0.1. An aliquot of each culture was collected at $t = 0$ h, $t = 3$ h, $t = 6$ h, $t = 9$ h, $t = 12$ h, and $t = 24$ h and the OD$_{600nm}$ was measured. For each time point, the cultures' aliquots were also serially diluted and plated on LB agar plates to measure CFU/mL or plated onto *E. coli* DH5α top lawns to evaluate the number of released phages (PFU/mL). For experiments in native conditions, Zorya II cells were challenged with phage phAvM.

## Membrane potential and membrane permeability analysis

For flow-cytometry assays, *E. coli* MT56 harboring pGM39 (empty vector) or the same vector encoding Zorya I, and Zorya II were grown in LB supplemented with kanamycin and 0.2% *L*-Rhamnose for 2 h. Cells were normalized to an OD$_{600nm}$ of 1 and stained with DiBAC$_4$(3) (Bis-[1,3-Dibutylbarbituric Acid] Trimethine Oxonol; Thermo) at 10 μM. Stained samples were incubated for 10 min in the dark and subsequently washed with fresh LB. Cells were analyzed in a FACS LRS Fortessa equipped with a 488 nm laser (Becton Dickinson), using thresholds on the side and forward scatter to exclude electronic noise. Bacterial cells were selected using side scatter (SSC-A) vs forward scatter (FSC-A). For DiBAC$_4$(3) detection, the Alexa 488 channel (Ex 488 nm, Em 530/30 nm) was used. Analysis was performed using FlowJo v10.4.2 (Treestar Inc.). As a control for depolarization, cells were treated with polymyxin B (5 μg/mL) at 37 °C for 30 min prior to staining. PMB-treated cells were used to define the DiBAC$_4$(3)-positive quadrant.

For kinetic measurements of DiBAC$_4$(3) fluorescence during phage infection *E. coli* MT56 harboring pGM39 (empty vector) or the same vector encoding Zorya I, Zorya II or their mutants were grown in LB supplemented with kanamycin and 0.2% *L*-Rhamnose to an OD$_{600nm}$ of ~0.4. ϕAlma and ϕMak were added at a MOI of 1 for evaluation of Zorya I and Zorya II activity, respectively. DiBAC$_4$(3) was added to a final concentration of 250 nM and fluorescence measurements were performed using a 96-well optical-bottom black plate and TECAN infinite nano M+ Microplate reader, with an excitation wavelength of 488 nm and emission wavelength of 530 nm for DiBAC$_4$(3).

For kinetic measurements of propidium iodide (PI) fluorescence during phage infection, the same growth conditions as for DiBAC$_4$(3) were used. PI was added at a final concentration of 200 μM and fluorescent measurements were performed with the use of a 96-well optical-bottom black plate and TECAN infinite nano M+ Microplate reader, with an excitation wavelength of 544 nm and emission of 612 nm. As a control for both PI and DiBAC$_4$(3), cells were treated with PMB as above.

## CellTiter Blue metabolic assay

To assess the mechanism of Zorya I and Zorya II-expressing cells that are metabolically active, the CellTiter Blue stain (Promega) was used, following manufacturer instructions. Briefly, *E. coli* MT56 harboring pGM39 (empty vector) or the same vector encoding Zorya I, and Zorya II were grown in LB supplemented with kanamycin and 0.2% *L*-Rhamnose for 2 h. A volume of 90 μL of each culture was added to each well of a 96-well optical-bottom black plate. Ten microliter of the CellTiter Blue dye was then added to each well and the fluorescence at 560/590 nm was recorded with a TECAN infinite nano M+ Microplate reader.

As a positive control, *E. coli* Mt56 carrying empty vectors were incubated for 10 min at 100 °C.

## Fluorescence microscopy

Overnight cultures (5 mL) were diluted into 25 mL LB containing 0.2% *L*-Rhamnose and 100 μg/mL kanamycin and grown for 2 h. 200 μL of each culture were collected at timepoints $t = 0$ h, $t = 1$ h, and $t = 2$ h and stained with 4′,6-diamidino-2-phenylindole (DAPI) at a final concentration of 5 μg/mL. Cells mixed with DAPI were incubated at 37 °C for 15 min and then 1 μL of each culture was transferred on a microscope slide with a pad of 1% UltraPure agarose (Invitrogen) in H$_2$O. Images were collected on a Zeiss LSM980 Microscope equipped with Widefield Camera Axiocam 705 mono and Light Source Colibri 5 Type RGB-UV-4-channel fluorescence light source with integrated control unit and a Plan Apochromat 63x objective.

For quantification of DAPI fluorescence, individual cells were identified from thresholded brightfield images and converted to the region of interest (ROI) using Fiji. Within each image, only cells in focus were considered. Extremely elongated cells had to be excluded from the analysis as it was not possible to threshold them as single cells in Fiji.

Fluorescence images were background-subtracted and ROIs were used to measure the integrated density (sum of pixel values over the whole cell area) of the DAPI fluorescence signals. Data were plotted as a swarm plot on GraphPad Prism 9.

## Electrophoresis-based assay for detection of genomic DNA lesions (EAsy-GeL)

Detection of DNA damage on purified genomic DNA was performed as follows: *E. coli* MT56 harboring pGM39 (empty vector) or the same vector encoding Zorya I, Zorya II, or their mutants were grown for 2 h in LB supplemented with 0.2% *L*-Rhamnose[39]. For experiments in native conditions, *E. coli* MT56 harbouring empty vector (pSUPROM) or the same vector encoding Zorya II or its mutants were grown in LB supplemented with Kan for 2 h.

Genomic DNA was extracted with a Monarch genomic DNA extraction kit (NEB) following manufacturer instructions. DNA was eluted in a neutral buffer (100 mM Tris-HCl pH 7.5 1 mM EDTA) and treated with RNAse A for 15 min at 37 °C. For double-stranded breaks, 300 ng of genomic DNA resuspended in this neutral buffer was analyzed on 0.8% agarose gels for assessment of double-stranded breaks (DSB).

The presence of alkaline unwinding-sensitive sites (AU-SSs), such as single-stranded (SSB) was investigated with alkaline/neutral treatment for evaluation. Briefly, 3 μL of 1 M Na$_2$HPO$_4$ pH 1.85 was added to 20 μL of a neutral buffer containing 300 ng of genomic DNA. The solution was homogenized by pipetting and then 9 μL of 0.1 M HCl were added. The solution was homogenized again before incubating in ice for 4 min. Loading dye was added and DNA was analyzed on a 0.8% agarose gel.

For experiments during phage infection, cells were grown up to OD$_{600nm}$ = 0.6 and infected with ϕAlma (for Zorya I), ϕT7 (for Zorya II) or ϕphAvM (for Zorya II_native). Cells were then recovered after the first burst event and processed as detailed above.

## Purification of ZorAB complexes

*Shewanella* sp. strain ANA-3 (type I) and *S. kujiense* (type II) ZorAB were expressed in *E. coli* MT56 as a single operon from a pT12 vector encoding a C-terminal Twin-Strep tag. All purification steps were carried out at 4 °C. Briefly, cells were grown at 37 °C for 15 h in a terrific broth medium containing kanamycin (50 μg/mL) and *L*-Rhamnose monohydrate (0.1% *w/v*) and then collected by centrifugation at 4000×*g*. Cell pellets were resuspended in Tris-buffered saline (TBS) (100 mM of Tris, 150 mM of NaCl, 1 mM of EDTA, pH 8.0) plus 30 μg/mL of DNase I and 400 μg/mL of lysozyme for 30 min before passage through an EmulsiFlex-C3 homogenizer (Avestin) at 15,000 psi. Unbroken cells were removed by centrifugation at 24,000×*g* for 20 min. The supernatant was recovered and total membranes were

collected by centrifugation at 200,000×g for 1.5 h. Membranes were resuspended in TBS and solubilized by incubation with 1% (w/v) lauryl maltose neopentyl glycol (LMNG; Anatrace) for 1 h. Insoluble material was removed by centrifugation at 100,000×g for 30 min. Solubilized membranes were then applied to Strep-Tactin XT 4 flow cartridges (IBA) pre-equilibrated in TBS. The resin was washed with 10 column volumes of TBS containing 0.02% (w/v) LMNG and proteins were eluted in 5 column volumes of TBS supplemented with 0.02% (w/v) LMNG and 50 mM of D-biotin (IBA). Eluates were concentrated using a 100-kDa molecular weight cut-off (MWCO) Vivaspin 6 (GE Healthcare) centrifugal filter unit and injected onto a HiLoad Superose 6 16/600 pg size-exclusion column (GE Healthcare) pre-equilibrated in TBS plus 0.02% (w/v) LMNG. Peak fractions were collected and either diluted in TBS plus 0.02% (w/v) LMNG (*S. kujiense*) or concentrated (*Shewanella* sp. ANA-3) using a 100-kDa MWCO Vivaspin 500 (GE Healthcare) centrifugal filter unit (Supplementary Fig. 8a, b).

### Cryo-EM sample preparation and imaging

Four microliters of *Shewanella* sp. strain ANA-3 (type I) and *S. kujiense* (type II) ZorAB sample at an $A_{280nm}$ of 0.25 and 2.3, respectively, were applied onto a glow-discharged (30 s, 25 mA) 300-mesh R1.2/1.3 Quantifoil Au grids. Grids were blotted for 2 s in 100% humidity at 8 °C and plunged frozen in liquid ethane using a Vitrobot Mark IV (Thermo Fisher Scientific).

Data were collected in a counted mode in EER format on a CFEG-equipped Titan Krios G4 (Thermo Fisher Scientific) operating at 300 kV with a Selectris X imaging filter (Thermo Fisher Scientific) with a slit width of 10 e⁻V and Falcon 4 direct detection camera (Thermo Fisher Scientific) at 165,000x magnification, with a physical pixel size of 0.723 Å. Movies were collected at a total dose of 54.0 e⁻/Å² (ZorAB type I) or 57.0 e⁻/Å² (ZorAB type II), both fractionated to 1 e⁻/Å² per frame.

### Cryo-EM data processing

Patched (20 × 20) motion correction, CTF parameter estimation, particle picking, extraction, and initial 2D classification were performed in SIMPLE 3.0[47]. All downstream processing was carried out in Relion 3.1 or cryosparc 3.3.1[48,49]. Gold-standard Fourier shell correlations (FSCs) using the 0.143 criterion were calculated within cryoSPARC and local resolution estimations were calculated within Relion. For *Shewanella sp. strain ANA-3* type I ZorAB (Supplementary Fig. 8), 1,252,092 particles were selected after one round of reference-free 2D classification (k = 400) within cryoSPARC. A soft circular mask of 130 Å in diameter was used in the previous step to ensure the centering of the core. Four volumes were generated from the 2D-cleaned particles after multi-class ab initio reconstruction using a maximum resolution cutoff of 12 Å. Only one volume showed features consistent with the 2D averages. This volume was lowpass-filtered to 20 Å and used as input for a 4-class heterogeneous refinement, resulting in a map with strong structural features. Particles (766,582) from this volume class were selected and non-uniform refined against their corresponding volume lowpass-filtered to 30 Å, generating a 3.0 Å map. Bayesian polishing followed by an additional round of 2D classification further truncated the particle subset to 587,313 particles. These particles were then used as input for non-uniform refinement against the same reference as the previous (lowpass-filtered to 30 Å) to generate a 2.2 Å volume. Per-particle defocus refinement and per-group CTF refinement were attempted but did not improve map quality. Local refinement of the peptidoglycan-binding domains using a soft mask encompassing these domains was performed, resulting in a 2.4 Å map with improved interpretability. A composite map was then generated using both globally B-factor sharpened maps, using the combine_focused_maps module in PHENIX[50]. For *Sulfuricurvum kujiense* type II ZorAB (Supplementary Fig. 9) 2,360,094 particles were selected after initial 2D classification (k = 500) using cluster2D_stream in SIMPLE. These

particles were further subjected to two consecutive rounds of reference-free 2D classification in cryoSPARC (k = 200 each). 2D-cleaned particles (768,077) were then subjected to multi-class ab initio reconstruction, using a maximum resolution cutoff of 5 Å, generating three volumes. Particles (369,395) from the most populated and featureful volume were selected and non-uniform refined against their corresponding volume lowpass-filtered to 8 Å, generating a 3.0 Å map. Bayesian polishing followed by an additional round of 2D classification further truncated the particle subset to 366,883 particles. These particles were then used as input for non-uniform refinement against the same reference as the previous (lowpass-filtered to 8 Å) to generate a 2.8 Å volume. Per-particle defocus refinement and per-group CTF refinement were attempted but did not improve map quality. To improve density for the cytoplasmic domains of ZorA, particle subtraction of the peptidoglycan-binding domains of ZorB and partial TM helices of ZorAB was performed on particles belonging to the consensus 2.8 Å volume followed by reference-free 2D classification. Selected particles were subjected to multi-class ab initio (k = 3) using a maximum resolution cutoff of 5 Å, generating two volumes with clear helical density for the ZorA cytoplasmic extensions. While the transmembrane helices for both volumes were superimposable, the cytoplasmic extensions were tilted differently with respect to the membrane. Independent non-uniform refinement of particles from both volumes, using an initial lowpass filter of 8 Å, yielded 5.5 and 5.6 Å volumes. Particles belonging to the two classes with the strongest helical density in the cytoplasmic domains were reverted to the original particles and independently non-uniform refined against their corresponding 8 Å lowpass-filtered volumes as references, resulting in 3.1 Å and 3.3 Å maps. Both volumes aligned with a correlation of 0.97 using the fitmap command of ChimeraX[51], demonstrating that the core and peptidoglycan-binding domains of both maps were equivalent despite the difference in the tilt angle of the ZorA cytoplasmic extensions with respect to the core. Local refinement of the strongest class, using a soft mask encompassing partial TM helices and the cytoplasmic extension of ZorA, yielded a 4.9 Å volume with clearly defined but partial cytoplasmic helices.

### Model building, refinement, and interpretation

Atomic models for *Shewanella* sp. strain ANA-3 and *Sulfuricurvum kujiense* ZorAB were built de novo using Coot v.0.9.8.3[52], with the cytoplasmic domains of *Shewanella* sp. strain ANA-3 ZorA guided by an AlphaFold[53] model. Multiple rounds of rebuilding in both the unsharpened and global B-factor sharpened maps followed by real-space refinement in PHENIX[4] using rotamer and Ramachandran restraints yielded the final models described in Table 1. Waters were placed into the *Shewanella* model using Douse within PHENIX[4]. All models were validated using MolProbity within PHENIX[54]. The placement of sodium ions in the *Shewanella* model was validated using CheckMyMetal[55].

Conservation analysis was carried out using AL2CO within ChimeraX[51,56]. ZorA cytoplasmic rod models (residues 237–696 of *Shewanella* sp. strain ANA-3; residues 110–378 of *S. kujiense*) were generated using AlphaFold-multimer[57]. Overlays between ZorAB and *C. sporogenes* MotAB (PDB 8UCS) were performed with the core inner helices of ZorA/MotA and the transmembrane helix of ZorB/MotB using the superpose module within CCP4[58]. Structural alignments of the ZorB peptidoglycan binding domain of *S. kujiense* and *H. influenza* peptidoglycan-associated lipoprotein (Pal) to the *Shewanella* sp. strain ANA-3 ZorB peptidoglycan-binding domain were performed using SSM superpose in Coot[52]. Figures were prepared using UCSF ChimeraX v.1.7[51] (Supplementary Fig. 10).

### Protein expression and purification

For large-scale expression of ZorE-Strep for biochemistry, *E. coli* MT56 was transformed with pGM29. Single colonies were then used to inoculate 150 mL Terrific Broth (Melford) supplemented with Kn for

overnight growth at 37 °C with 180 rpm shaking. Starter cultures were re-seeded 1:100 $v/v$ into each of $12 \times 2$ L baffled flasks containing 1 L Terrific Broth supplemented with Km and were subsequently incubated at 37 °C with 150 rpm shaking until reaching an $OD_{600nm}$ of 0.4. Flasks were then supplemented with L-Rhamnose to a final concentration of 0.2% $w/v$, and incubated for a further 4 h at 37 °C with 150 rpm shaking. Cells were harvested by centrifugation at 4200×$g$ for 30 min at 4 °C, then serially resuspended in ice-cold buffer A (50 mM Tris HCl pH 8.0, 150 mM NaCl, 10% glycerol). Resuspended cells were disrupted by sonication (40% amplitude, 10 s pulses with 20 s rest, 3 min total pulse) and clarified by centrifugation at 45,000×$g$ for 50 min at 4 °C. Clarified cell lysate was transferred to a chilled glass beaker on ice and applied to a 5 mL StrepTrap HP column (Cytiva) pre-equilibrated in buffer A. The StrepTrap column was then washed with 50 mL buffer A. Bound proteins were then eluted with 50 mL of buffer B (50 mM Tris HCl pH 8.0, 150 mM NaCl, 2.5 mM desthiobiotin, 10% glycerol). The eluate was subsequently concentrated by centrifugation using a 10 kDa MWCO Vivaspin concentrator (Sartorius) and the concentrated protein sample was then applied to a Superdex 75 increase 10/300GL (S-75; Cytiva) pre-equilibrated in sizing buffer (50 mM Tris HCl pH 7.9, 500 mM KCl, 10% glycerol). The resulting peak was centrifugally concentrated to ~1 mg/mL, snap-frozen in liquid nitrogen in aliquots ready for use, and stored at −80 °C. ZorE (1 µg) was resolved on a 4–20% ($v/v$) polyacrylamide Mini-PROTEAN TGX precast gel for 15 min at 300 V.

## Mass photometry

Solution-phase mass determination of ZorE-Strep was performed using the TwoMP (Refeyn) mass photometer. Samples were first diluted ~1000-fold in PBS buffer A25. Experimental data were obtained in the form of mass photometry videos recorded for 1 min using the AcquireMP v2.5 software (Refeyn) on precleaned, poly-lysine-treated high-sensitivity microscope slides. A mass calibration was done using thyroglobulin, aldolase, and conalbumin from the calibration kits (Cytiva). The experimental data were then fit to this calibration, and graphs were generated using the DiscoverMP v2.5 software (Refeyn).

## Nuclease assays

To test the efficiency of ZorE nuclease activity we used the pSG483 plasmid. This plasmid carries a unique Nb.Bpu10I site, for nicking of the plasmid. The nicking reaction was set up by adding 500 ng of pSG483 and 15 Nb.Bpu10I (Thermo) in a final volume of 300 µL. The nicking reaction was incubated for 4 h at 37 °C and inactivated at 80 °C for 20 min. Relaxed DNA was obtained by adding ATP and T4 ligase (NEB) to the mix for 1 h at room temperature. BamHI digestion for 1 h at 37 °C was used to obtain linear pSG483.

For titration experiments, 0 nM, 48 nM, 96 nM, 192 nM, 384 nM, 768 nM, and 1536 nM of purified ZorE were incubated with 6 nM of pSG483 plasmid or 200 ng of $E. coli$, φT7or φMak gDNA. Samples were incubated for 60 min in the presence of 5 mM $Mg^{2+}$ at 37 °C.

To test the activity of ZorE in the presence of various metals, ZorE (768 nM) was incubated with supercoiled pSG483 (6 nM) at 37 °C for 60 min in the presence of 5 mM, 25 mM, and 50 mM $Mg^{2+}$, $Mn^{2+}$, $Ca^{2+}$, $Zn^{2+}$, and $Li^{+}$.

To test ZorE specificity for DNA topoisomers, ZorE (768 nM) was incubated with supercoiled, relaxed, linear, and nicked plasmid pSG483 (6 nM) at 37 °C for 0 to 120 min with 5 mM MgOAc.

All reactions were stopped by the addition of EDTA and SDS and products were analyzed by gel electrophoresis in a 1× TAE, 1.4% agarose gel, post-stained with ethidium bromide.

## Purification of ZorB PG-binding domains

ZorB I$_{165-287}$ (from $S. marcescens$ ATCC 274) and ZorB II$_{115-235}$ (from $E. coli$ ATCC8739) or their point mutations as detailed in Fig. 1 were cloned in a pT12-based plasmid under the control of a L-Rhamnose-

inducible and in frame with a C-term twin-strep tag. Constructs were then transformed in $E. coli$ MT56. Strains were inoculated in 1 L terrific broth (Formedium) at a starting $OD_{600nm}$ of 0.05 and grown at 30 °C. When cells reached $OD_{600nm} = 0.6$, L-Rhamnose was added at a final concentration of 0.2%. Cells were then grown for 12 h at 16 °C and recovered by centrifugation 4000×$g$, 30 min. Recovered cells were resuspended in 10 mL of Buffer A (50 mM Tris-HCl pH 8, 150 mM NaCl) in the presence of cOmplete™ EDTA-free protease inhibitor (Merck) and lysed by sonication (cycles of 20 min on, 20 min off, amplitude 70%). The lysate was cleared by centrifugation (14,000×$g$, 45 min, 4 °C), filtered through a 0.45 µm filter, and added to 1 mL column-volume of Strep-Tactin™XT Sepharose resin (IBA Lifesciences) pre-equilibrated with Buffer A. The unbound lysate was removed by centrifugation for 2 min at 700×$g$. The resin was subsequently washed with 10-column volumes of Buffer A and elution was performed with 5-column volumes of Buffer B (100 mM Tris-HCl, 150 mM NaCl, 50 mM biotin, pH 8.0).

## PG-binding assay

Peptidoglycan isolation was performed using an adapted protocol based on previously described methods[37,59,60]. $E. coli$ Mt56 was grown in 2 L of LB until the late exponential phase (~$OD_{600nm} = 1$). Cells were collected and resuspended in 10 mL of buffer A (100 mM Tris pH 7, 500 mM NaCl). SDS was added to a final concentration of 6% and samples were boiled for 1 h. Samples were centrifuged at 80,000×$g$ for 10 min and SDS was removed by washing pellets 10× times with 5 mL of MilliQ water. Samples were resuspended in 20 mL of 100 mM Tris pH 7 and then treated with 15 µg/mL of DNase and 60 µg/mL RNase for 2 h at 37 °C. Next, 100 µg/mL of trypsin was added, and samples were incubated overnight at 37 °C.

The following day, EDTA and SDS were added to the sample at a final concentration of 10 mM and 1%, respectively. Samples were boiled for 20 min and then centrifuged at 80,000×$g$ for 1 hr. Pellets were washed 5× times with MilliQ water and finally resuspended 100 mM Tris, pH, 100 mM NaCl

For the PG-binding assay, ZorB I$_{165-287}$ and ZorB II$_{115-235}$ or their point mutants as detailed in Fig. 2e–h (75 mg) were mixed to PG in a fresh tube and incubated at 25 °C for 1 h on an end-over-end rotator at 10 rpm. Samples were centrifuged at 20,000×$g$ for 30 min and washed 3× times with 100 mM Tris, pH, and 100 mM NaCl. Following the third wash, the supernatant was retained for analysis on SDS-PAGE. The pellet was resuspended in 15 mL of 100 mM Tris, pH, 100 mM NaCl, and 5 mL of 4× Laemni buffer was added for analysis on SDS-PAGE.

## Subcellular fractionation

For separation of cytoplasm and total membrane fractions, $E. coli$ MT56 cultures as reported in Fig. 6 were grown to an $OD_{600nm} = 0.6$ in a final volume of 500 mL of LB. φphAvM was added to an MOI of 0.1 and cells were grown until the first burst event. Cells were recovered by centrifugation and resuspended in 1 mL of buffer A (50 mM Tris HCl pH 8.0, 150 mM NaCl). Samples were sonicated (40% amplitude, 10 s pulses with 20 s rest, 3 min total pulse) and debris was removed by centrifugation (13,000×$g$, 20 min, 4 °C). The cleared supernatant was subjected to ultracentrifugation (80,000×$g$, 30 min, 4 °C), and the resulting supernatant, representing the cytoplasm, was mixed with Laemni buffer for analysis on SDS-PAGE. The pellet was resuspended in 500 mL of buffer A and an aliquot, representing the total membrane fraction, was mixed with Laemni buffer for analysis on SDS-PAGE.

Samples were then subjected to immunoblot analysis. ZorE-His was detected using an Anti-His monoclonal antibody (1:6000, Invitrogen) and ZorB-Strep using an anti-Strep monoclonal primary antibody (1:10,000, Qiagen), both with an HRP-conjugated anti-Mouse secondary antibody (1:10,000, Biorad). For the cytoplasmic control, an anti-CsrA polyclonal antibody (1:2000, Cliniscences) was used, whereas, for the membrane control, an anti-OmpC polyclonal antibody (1:2000, Cliniscences) was used. For both, detection was obtained

with an HRP-conjugated anti-rabbit secondary antibody (1:10,000, Biorad).

## In silico analysis

HMMbuild from the HHMER suite (v 3.3.2) was used to build Hidden Markov Models for ZorA I, ZorB I, ZorA II, and ZorB II. Models were built based on ZorA and ZorB homologs first identified by Doron et al. [2], Hmmsearch from the HHMER suite (v 3.3.2) was then used to query a local database of bacterial proteins. Protein alignments of ZorA and ZorB homologs were generated with MUSCLE (v3.8.1551)[61]. Alignments were trimmed with TrimAL and concatenated with seqkit[62,63]. The maximum likelihood phylogenetic tree was built using IQTree-2 v2.3.6 and annotated on iTOL[64,65].

To identify bacterial genomes that contain both Zorya I and Zorya II operons cblaster v 1.3.18 was used[66]. Filters used were minimum identity (-mi) = 30%, minimum coverage(-mc) = 60% and minimum hits in a cluster (-mh) = 6.

## Reporting summary

Further information on research design is available in the Nature Portfolio Reporting Summary linked to this article.

## Data availability

Cryo-EM volumes and atomic models have been deposited to the EMDB and PDB (accession codes EMD-43560, EMD-43561, EMD-43562, EMD-43563, 8VVN, 8VVI). ZorA and ZorB alignments and structure validation reports are available at https://github.com/GM110Z/Zorya-paper. DNA gels used for quantification of ZorE nickase activity are provided at https://doi.org/10.6084/m9.figshare.28319225. All the remaining data generated in this study and necessary for interpretation are provided in the Supplementary Information and/or Source Data file. Source data are provided with this paper.

## Code availability

All custom scripts used can be found at: https://github.com/GM110Z/Zorya-paper.

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

## Acknowledgements

This work was funded by a Wellcome Trust Sir Henry Wellcome Fellowship (218622/Z/19/Z) to G.M. An Intramural Research Program of the NIH to S.M.L, a Biotechnology and Biological Sciences Research Council Newcastle-Liverpool-Durham Doctoral Training Partnership studentship [grant number BB/T008695/1] to J.J.R., an Engineering and Physical Sciences Research Council Molecular Sciences for Medicine Center for Doctoral Training studentship [grant number EP/S022791/1] to M.J.G. and a Lister Institute Prize Fellowship to T.R.B. The authors wish to thank Prof Graham Stewart, Auron Pileri, Dr Rachel Butler, and Josie Butler for technical assistance. The authors also wish to thank Dr Abigail Kelly for her technical assistance in the Mass Photometry (Refeyn) data collection. For the purpose of open access, the authors have applied a CC BY public copyright license to any Author Accepted Manuscript version arising from this submission.

## Author contributions

G.M., J.C.D., and S.M.L. conceptualized the study. G.M., J.C.D., J.J.R., M.J.G., S.S., L.B., M.K., and Y.E. performed experimental work. G.M., J.C.D., T.R.B, T.P., and S.M.L analyzed data. G.M., J.C.D., and S.M.L. wrote the manuscript with contributions from all authors.

## Competing interests

The authors declare no competing interest.
