## [Transparent Peer Review file · Nature Communications]

Modularity of Zorya defence systems during phage inhibition

Corresponding Author: Dr Giuseppina Mariano

Version 0:

Reviewer comments:

Reviewer #1

(Remarks to the Author)

The authors have responded to the latest set of concerns I raised in review. Although I find some of their answers sufficient, the key concerns have not been adequately addressed. To recap:

1) Does the association of ZorB with peptidoglycan change during phage infection? This is crucial to any model of how this system senses phage. The authors argued that it's too difficult to measure.

2) Is ZorE recruited to ZorAB during phage infection? It was stated that an experiment to test this was tried but didn't work because of cell death. However, during an initial round of infection, cells are not dead so I don't understand the authors' argument here. This seems like a straightforward experiment and is crucial to understanding how the system works.

3) I raised concerns about overexpressing the systems at hand and the artifacts that can arise. The authors argue that artifacts from overexpression are not concerning because other labs also examine defense systems following overexpression. I don't agree with this logic – two wrongs don't make a right. There are also important, nuanced differences between modest overexpression, e.g. plasmid v. chromosome with native promoters, and substantial overexpression, e.g. strong, inducible promoters. I do agree with the authors' assertion that overexpression-based experiments can provide useful hints, but I think they still need to be validated in the context of native promoters/regulation.

4) Most importantly: the authors did not adequately address the concerns raised about the bold claims made about Zorya II and retron IIA acting as Abi alone but as direct defense together. Looking at the data in ED Fig. 9 and to reiterate a point from the prior review: in the presence of both systems, the CFU counts do not drop as much as with either system alone, but they still drop substantially, i.e. they drop 1000-fold (!) after 3 hrs. If this were a case of direct defense, e.g. an RM system, the CFUs would never drop. As also noted last time, the effects seen in ED Fig. 9 involve artificial expression, so the relevance of the observations to a native situation remain unclear. I acknowledge that Fig. 6b suggests a possible synergy between Zorya II and retron IIA, but I simply do not agree with the conclusion that this is direct defense. The authors have not presented compelling or sufficient evidence in favor of this conclusion.

5) The basis of the synergy also remains unclear. There is an IP-MS experiment to suggest that the retron effector may interact (directly or indirectly) with the Zorya system but this interaction is still not validated. Does Zorya recruit the retron following phage infection (as also asked about for ZorE above)? Is the interaction direct? If recruitment is ablated, does synergy go away? Does the interaction somehow promote activity of the retron effector? What is the effector's activity anyway? This part of the manuscript continues to be severely underdeveloped. I recognize that some aspects of this go beyond the scope of this study, but as it stands now the manuscript at hand is insufficient to conclude much about the synergy detected. Moreover, as emphasized in the last review, the effect size here is quite small. Even with the artificial levels being used, the synergy is only ~10-fold. The authors retorted that as long as it's statistically significant it must be biologically significant. But statistical and biological significance are not equivalent!

Reviewer #4

(Remarks to the Author)

In this revised manuscript, the authors provide robust and comprehensive biochemical and functional data to support their structures of ZorAB complexes. This is stellar work, and I strongly recommend its publication. There are a couple of very minor points which would improve the clarity of the manuscript, which can be easily implemented. This is a great piece of work and represents a significant advancement for the field, and the new insights into synergy with the Retron IIA system are compelling.

1. Please add domain annotations to the schematic representations in Fig 1a.
2. It might also be useful to recolor these schematics - I appreciate the color gradients, but having ZorA and ZorB consistent colors between Fig1a and Fig1b would be helpful (i.e. purple and blue). If the authors could add labels in panels b and c to distinguish the ZorA and ZorB domains, then it would be even clearer.
3. In Fig 5, it might be useful to re-arrange the figures. If panel D is quantification of panel C, then it would be better for them to be side-by-side.
4. It might be useful to include AlphaFold models of ZorC-E in the supplementary materials.

Jack Bravo (with Balwina Koopal)

Reviewer #5

(Remarks to the Author)

Version 1:

Reviewer comments:

Reviewer #1

(Remarks to the Author)

I have no further issues for the authors to address. I think removing the synergy section/data was a wise decision.

Reviewer #1 (Remarks to the Author):
The authors have responded to the latest set of concerns I raised in review. Although I find some of their answers sufficient, the key concerns have not been adequately addressed. To recap:

1) Does the association of ZorB with peptidoglycan change during phage infection? This is crucial to any model of how this system senses phage. The authors argued that it's too difficult to measure.

We appreciate that measuring changes in the ability of ZorB to bind PG *in vivo* would be ideal. Our experiments demonstrate that ZorB proteins have the potential to bind PG *in vitro*. *In vivo* and in the absence of phage infection, we envision that the lack of binding may be attributed to the spatial separation between PG and the ZorAB complex in a resting state. It is plausible that phage-induced membrane puncturing pushes the PG inward, bringing it into contact with ZorB. This interaction could enable ZorB to bind PG and trigger downstream changes that recruit ZorE. However, observing these dynamics *in vivo* under microscopic conditions is challenging and not trivial to achieve.

2) Is ZorE recruited to ZorAB during phage infection? It was stated that an experiment to test this was tried but didn't work because of cell death. However, during an initial round of infection, cells are not dead so I don't understand the authors' argument here. This seems like a straightforward experiment and is crucial to understanding how the system works.

We thank the reviewer for their insightful comment. In response, we have repeated this experiment using a construct expressing Zorya II (with ZorE-His and ZorB-Strep) or its mutants under the control of a native promoter. The new data, presented in Figure 6k, demonstrate specific recruitment during phage (phAvM) infection. To perform these experiments, we utilized newly isolated environmental phages to observe robust phenotypes associated with natively expressed Zorya II. This approach was necessary because Zorya II-mediated protection was insufficient against the T7 phage under native expression conditions (Figure 4).

3) I raised concerns about overexpressing the systems at hand and the artifacts that can arise. The authors argue that artifacts from overexpression are not concerning because other labs also examine defense systems following overexpression. I don't agree with this logic – two wrongs don't make a right. There are also important, nuanced differences between modest overexpression, e.g. plasmid v. chromosome with native promoters, and substantial overexpression, e.g. strong, inducible promoters. I do agree with the authors' assertion that overexpression-based experiments can provide useful hints, but I think they still need to be validated in the context of native promoters/regulation.

We thank the reviewer for their insightful comment. In response, we have now provided evidence that Zorya II-mediated protection still falls within the broader definition of population-wide immunity. Specifically, we show that Zorya II exerts a negative effect on bacterial growth/fitness through bacteriostasis, ultimately thwarting phage infection (Figure 5). Furthermore, we demonstrate that under native expression conditions, ZorE is recruited specifically to the site of infection during phage attack, a process mediated by ZorAB (Figure 6k). Finally, using the Easy-gel assay, we show that under

native conditions and in the absence of infection, no damage is detected in total DNA. Importantly, during infection, ZorE retains its predominant nickase activity (Figure 6j).

These findings validate the conclusions drawn from our overexpression studies, confirming that the observed defense mechanisms hold true under native promoters and regulation. While Zorya II overexpression data suggest a population-wide immunity mechanism that could involve cell death rather, the data obtained with native expression point towards bacteriostasis. This difference could indeed arise from overexpression. Nonetheless, within the broader concept of abortive infection—as a mechanism that reduces bacterial fitness without necessarily requiring cell death—our data support the conclusion that the Zorya II-mediated mechanism encompasses population-wide immunity or an abortive infection phenotype.

We hope this addresses the reviewer's concerns regarding potential artifacts arising from overexpression-based experiments and demonstrates the robustness of our results under physiologically relevant conditions.

We further provide several references of literature showing that population-wide immunity can include mechanism such as stasis or persistence:

<https://journals.asm.org/doi/10.1128/spectrum.03388-23>

<https://www.biorxiv.org/content/10.1101/2024.11.23.624991v1>

<https://www.sciencedirect.com/science/article/pii/S0092867420313064>

https://academic.oup.com/nar/article/52/8/4723/7642066#google_vignette

4) Most importantly: the authors did not adequately address the concerns raised about the bold claims made about Zorya II and retron IIA acting as Abi alone but as direct defense together. Looking at the data in ED Fig. 9 and to reiterate a point from the prior review: in the presence of both systems, the CFU counts do not drop as much as with either system alone, but they still drop substantially, i.e. they drop 1000-fold (!) after 3 hrs. If this were a case of direct defense, e.g. an RM system, the CFUs would never drop. As also noted last time, the effects seen in ED Fig. 9 involve artificial expression, so the relevance of the observations to a native situation remain unclear. I acknowledge that Fig. 6b suggests a possible synergy between Zorya II and retron IIA, but I simply do not agree with the conclusion that this is direct defense. The authors have not presented compelling or sufficient evidence in favor of this conclusion.

We appreciate that the reviewer remains unconvinced and have therefore removed these data from the manuscript. We plan to investigate the mechanistic details of this synergy further and address them in a separate study. As part of removing the synergy data, we also excluded experiments involving the phages Mak/BAM. These experiments showed protection in the absence of ZorE, a phenotype we cannot fully explain without incorporating the Zorya-Retron experiments. To avoid leaving unresolved questions in this manuscript, we decided to omit these results and address them in future work.

5) The basis of the synergy also remains unclear. There is an IP-MS experiment to suggest that the retron effector may interact (directly or indirectly) with the Zorya system but this interaction is still not validated. Does Zorya recruit the retron following phage infection (as also asked about for ZorE above)? Is the interaction direct? If

recruitment is ablated, does synergy go away? Does the interaction somehow promote activity of the retron effector? What is the effector's activity anyway? This part of the manuscript continues to be severely underdeveloped. I recognize that some aspects of this go beyond the scope of this study, but as it stands now the manuscript at hand is insufficient to conclude much about the synergy detected. Moreover, as emphasized in the last review, the effect size here is quite small. Even with the artificial levels being used, the synergy is only ~10-fold. The authors retorted that as long as it's statistically significant it must be biologically significant. But statistical and biological significance are not equivalent!

See answer to point 4.

Reviewer #4 (Remarks to the Author):

In this revised manuscript, the authors provide robust and comprehensive biochemical and functional data to support their structures of ZorAB complexes. This is stellar work, and I strongly recommend its publication. There are a couple of very minor points which would improve the clarity of the manuscript, which can be easily implemented. This is a great piece of work and represents a significant advancement for the field, and the new insights into synergy with the Retron IIA system are compelling.

We appreciate the reviewer's positive feedback on our work and specifically on the data regarding the synergy between Zorya and RetronIIA. However, in response to other reviewers' concerns, we decided to remove this analysis from the current study. We plan to explore this intriguing interaction in more detail in future work.

1. Please add domain annotations to the schematic representations in Fig 1a. These have now been added as a separate panel Fig1.b. The separate panel was necessary to be able to cover the diversity of domains of each component of each Zorya subtype, whilst maintaining accessibility.

2. It might also be useful to recolor these schematics - I appreciate the color gradients, but having ZorA and ZorB consistent colors between Fig1a and Fig1b would be helpful (i.e. purple and blue). If the authors could add labels in panels b and c to distinguish the ZorA and ZorB domains, then it would be even clearer.

This is now edited.

3. In Fig 5, it might be useful to re-arrange the figures. If panel D is quantification of panel C, then it would be better for them to be side-by-side.

We thank the reviewer for their suggestion. We have now re-organised the Figure. We note that with the addition of new data, in response to Referee 1's comments, this has now become Figure 6.

4. It might be useful to include AlphaFold models of ZorC-E in the supplementary materials.

We thank the reviewer for their suggestion. AlphaFold models have now been added

as Supplementary Figure S1. Where relevant, alignments with structural homologues identified through Foldseek searches have also been included.

Jack Bravo (with Balwina Koopal)